# Learning in Compact Spaces with Approximately Normalized Transformer

**Jörg K.H. Franke**[1,2,3,4]       **Urs Spiegelhalter**[1]       **Marianna Nezhurina**[3,4,5]

**Jenia Jitsev**[3,4,5]       **Frank Hutter**[1,2,6]       **Michael Hefenbrock**[7]

[1]University of Freiburg   [2]ELLIS Institute Tübingen   [3]Open-Sci Collective
[4]LAION   [5]Jülich Supercomputing Centre (JSC)   [6]Prior Labs   [7]Perspix.ai

## Abstract

The successful training of deep neural networks requires addressing challenges such as overfitting, numerical instabilities leading to divergence, and increasing variance in the residual stream. A common solution is to apply regularization and normalization techniques that usually require tuning additional hyperparameters. An alternative is to force all parameters and representations to lie on a hypersphere. This removes the need for regularization and increases convergence speed, but comes with additional costs. In this work, we propose a more holistic, approximate normalization via simple scalar multiplications motivated by the tight concentration of the norms of high-dimensional random vectors. Additionally, instead of applying strict normalization for the parameters, we constrain their norms. These modifications remove the need for weight decay and learning rate warm-up as well, but do not increase the total number of normalization layers. Our experiments with transformer architectures show up to 40% faster convergence compared to GPT models with QK normalization, with only 3% additional runtime cost. When deriving scaling laws, we found that our method enables training with larger batch sizes while preserving the favorable scaling characteristics of classic GPT architectures.

## 1 Introduction

Normalization techniques, such as LayerNorm, are fundamental for stable and efficient Transformer training [1–4]. Loshchilov et al. [5] extends the concept and proposes the normalized Transformer (nGPT), where all latent residual representations and all parameters in the direction of the residual are normalized to lie on a hypersphere. We argue that the benefits of normalization come from two effects. Normalization prevents the representations on the residual stream from blowing up and requiring deeper layers to significantly amplify their output magnitudes. An effect observed by Sun et al. [6] and termed the "Curse of Depth" (see Figure 1). Additionally, normalization ensures a consistent input scale, which allows for the selection of a more suitable (global) learning rate.

These benefits drive us toward architectures that consistently apply normalization throughout the network, potentially normalizing all representations. Unfortunately, such excessive use of normalization increases training and, more importantly, inference times. To combat this problem, we introduce an approximate normalization technique for normalizing a vector $x$ via *normalizing factors* $\nu$ satisfying

$$\nu \approx (\|x\|_2)^{-1} \quad \text{so that} \quad \|\nu \cdot x\|_2 \approx 1.$$

39th Conference on Neural Information Processing Systems (NeurIPS 2025).

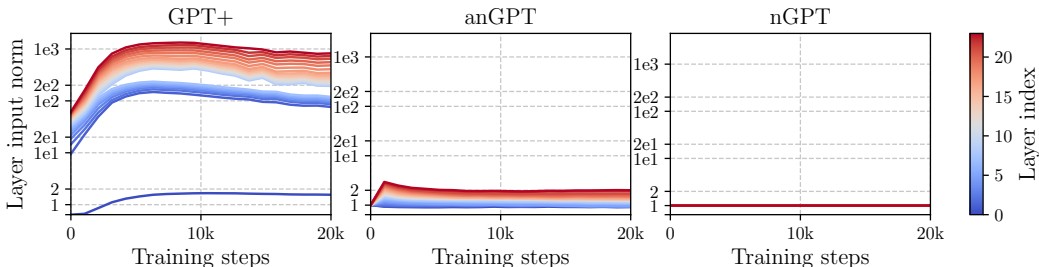

Figure 1: The input norm on log scale for each layer as a function of training a $0.5B$ model on $10B$ tokens. Deeper layers obtain a higher input norm in the classical GPT. While nGPT completely eliminates this "Curse of Depth", anGPT effectively mitigates it.

These normalizing factors are enabled by the concentration of measure phenomenon in high-dimensional spaces, which may be perceived as a "blessing of dimensionality". The augmentation of each operation with (approximate) normalizations, along with other modifications like bounding the norm of the input dimension of each linear map, forms the basis of our proposed approximately normalized Transformer (anTransformer). When applied to pretraining large language models, we adapt the GPT architecture to an approximately normalized GPT (anGPT) architecture without the need for additional normalization layers. Our approach *does not require weight decay nor learning rate warm-up*, effectively reducing the number of hyperparameters in training. Compared to a vanilla GPT architecture, anGPT achieves up to $40\%$ convergence speedup. Measurements show less than $3\%$ larger training step runtime, while expecting further reduction for inference times by subsuming normalization factors into the model parameters.

Our core contributions can be summarized as follows:

- We motivate the benefits of normalization in Section 3.
- The proposed approximately normalized transformer (anTransformer) in Section 4 displays faster convergence and fewer hyperparameters at a minor increase in training time.
- Section 5 presents extensive experimental evaluations:
  - Hyperparameter scaling trends are derived across multiple model sizes.
  - Results demonstrate over 40% convergence speedup compared to GPT models with QK normalization and outperform or perform on par with nGPT.
  - Compute-dependent scaling laws reveal scaling behavior matching GPT.

## 2   Related Work

Normalization techniques for deep neural networks have evolved significantly, from BatchNorm [7], which standardizes across batch dimensions but struggled with sequential data, to LayerNorm [1], which computes statistics across feature dimensions independently for each sample, making it ideal for transformers. The original Transformer [2] uses a Post-Norm configuration (LayerNorm after residual connections), requiring careful learning rate warm-up to prevent gradient explosion, while Pre-Norm architectures (normalization before operations) offer more stable training dynamics with higher potential learning rates [3, 4]. RMSNorm [8] further refined LayerNorm by eliminating the mean-centering step, delivering runtime improvements while maintaining performance in state-of-the-art LLMs [9, 10]. The normalized Transformer (nGPT) uses the normalized representation and parameter space to ensure that input tokens "travel" on the surface of a hypersphere, with each layer contributing a displacement towards target output predictions. These modifications render weight decay and learning rate warm-up unnecessary as vector magnitudes are explicitly controlled [5]. Empirically, nGPT demonstrates convergence speedups and reduces the number of training steps required to achieve equivalent accuracy. However, these improvements come with computational overhead due to additional normalization layers. Further related work is discussed in Appendix A.

# 3 Preliminaries

Normalized representations are known to stabilize training and accelerate convergence [3–5]. We hypothesize that this stems primarily from two effects. First, by ensuring each layer receives a normalized input, the input scale becomes consistent across the network. This uniformity can streamline learning and simplify the selection of a global learning rate. Second, as contributions to the residual stream accumulate in vanilla Transformer architectures, subsequent layers need to amplify their output to remain influential. This leads to a growing norm of the representation on the residual stream (see Figure 1), which may destabilize training. Both effects are elaborated below using toy examples.

## 3.1 Why does normalization influence optimization?

To better understand the role of the input scale in gradient descent on the learning rate, consider the problem $\min_{\boldsymbol{x}} \frac{1}{2}\boldsymbol{x}^\top \boldsymbol{\Lambda}\boldsymbol{x}$ with $\boldsymbol{\Lambda} = \mathrm{diag}(\lambda_1, \ldots, \lambda_d)$ and $\lambda_i > 0$. Using a learning rate of $\alpha > 0$, the gradient descent update is given by

$$\boldsymbol{x} \leftarrow \boldsymbol{x} - \alpha\boldsymbol{\Lambda}\boldsymbol{x} = (\boldsymbol{I} - \alpha\boldsymbol{\Lambda})\boldsymbol{x}.$$

Each entry $x_i$ of $\boldsymbol{x}$ converges with a rate determined by the contraction factor $\rho_i(\alpha) = |1 - \alpha\lambda_i|$. To ensure convergence, we require $\rho_i(\alpha) < 1$ with smaller values indicating faster convergence. Hence, the goal is to set $\alpha$ to minimize the maximum contraction factor $\max_i \rho_i(\alpha)$. The $\max_i \rho_i(\alpha)$ is either achieved for the smallest $\lambda_{\min}$ or the largest $\lambda_{\max}$. To achieve the same convergence speed for both, we require the optimal learning rate $\alpha^\star$ to satisfy

$$|1 - \alpha\lambda_{\max}| = |1 - \alpha\lambda_{\min}| \quad \Longrightarrow \quad \alpha^\star = \frac{2}{\lambda_{\max} + \lambda_{\min}} \quad \text{and} \quad \max_i \rho_i(\alpha^\star) = \frac{\kappa - 1}{\kappa + 1},$$

where $\kappa = \lambda_{\max}/\lambda_{\min}$ is the condition number of $\boldsymbol{\Lambda}$. One can see that the fastest contraction is realized if $\kappa = 1$, or equivalently, if $\lambda_{\max} = \lambda_{\min}$. Additionally, to ensure convergence, $\alpha$ must satisfy $0 < \alpha < 2/\lambda_{\max}$. If $\lambda_{\max} >> \lambda_{\min}$, we expect a slow convergence rate for the entry relating to $\lambda_{\min}$ as its contraction factor is bounded by the largest learning rate $\alpha$ that $\lambda_{\max}$ allows.

Now, consider a case where each column of $\boldsymbol{\Lambda}$ is normalized by multiplication of a diagonal preconditioner $\boldsymbol{P} = \mathrm{diag}(p_1, \ldots, p_d)$, with $p_j = (\|\boldsymbol{\lambda}_j\|_2)^{-1}$ where $\boldsymbol{\lambda}_j$ is the $j$-th column of $\boldsymbol{\Lambda}$. In this case, all $\lambda_i = 1$ and a learning rate can be picked that leads to the same convergence speed for all coordinates. This fact makes good learning rates more effective, regardless of how they are found (e.g, manual tuning, grid-search, or some sophisticated optimization). Consequently, normalization can serve as a (diagonal) preconditioner and improve the convergence speed of gradient-based learning methods.

While the example problem is simple, it may still provide some intuition about how normalization can help learning, namely, by allowing the selection of a well-working learning rate for all parameters. Even together with methods like Adam [11], normalization may provide benefits by improving the conditioning of the optimization landscape and yielding more stable gradient statistics for moment estimation. This can be seen by the gradual increase of the variance of the first moment, see Figure B.1.

## 3.2 Why does the norm of the residual connection increase?

Assume independent random vectors $\boldsymbol{h}_l$ with $\mathbb{E}[\boldsymbol{h}_l] = 0$ and $\|\boldsymbol{h}_l\|_2^2 = 1$, representing the contribution of each layer $l$ on the residual connection. For the $(L + 1)$-th layer to have an effective contribution to the residual state $\boldsymbol{h}_{\leq L} := \sum_{l=1}^{L} \boldsymbol{h}_l$, such as the ability to overwrite it, it has to have a magnitude similar to that of $\boldsymbol{h}_{\leq L}$. Specifically,

$$\mathbb{E}\big[\|\boldsymbol{h}_{\leq L}\|_2^2\big] = \mathbb{E}\left[\left\|\sum_{l=1}^{L} \boldsymbol{h}_l\right\|_2^2\right] = \sum_{l=1}^{L} \mathbb{E}[\|\boldsymbol{h}_l\|_2^2] + \sum_{l \neq l'} \mathbb{E}[\boldsymbol{h}_l]^\top \mathbb{E}[\boldsymbol{h}_{l'}] = \sum_{l}^{L} \mathbb{E}\big[\|\boldsymbol{h}_l|_2^2\big] = L.$$

Consequently, $\|\boldsymbol{h}_{L+1}\|_2 \approx \sqrt{L}$ is expected. Such effects lead to growing outputs on the hidden state that can also be observed experimentally, see Figure 1. Since the weights are subject to regularization, the large output scales are likely produced by the scaling factor $\gamma$ in the input norm of each block of

the transformer, see Figure C.1. Growing norms on the residual stream were also described in [6] and coined "Curse of Depth". As a fix, they proposed to scale the LayerNorm output of layer $l$ by $1/\sqrt{l}$. Alternatively, this growth can also be addressed by employing Post-Norms and keeping the residual connection normalized as in Loshchilov et al. [5].

# 4 Approximately Normalized Transformer

Due to the potential benefits of normalization, it is tempting to normalize the inputs for each primitive, namely, linear maps, activation functions, and residual updates. Unfortunately, such excessive use of normalizations might significantly influence the runtime [5]. However, to still keep the benefits of consistent normalization at a lower cost, this work explores an approach to replace normalization operations with cheaper, approximate computations.

## 4.1 Approximate Normalization

To reduce the overhead introduced by excessive normalization (in particular at inference time), the proposed method attempts to approximately normalize the representations in the architecture through *input independent normalization factors* $\nu$. Concretely, if for some vector $\boldsymbol{x}$, there exists some $\nu$, with

$$\nu \approx (\|\boldsymbol{x}\|_2)^{-1} \quad \text{so that} \quad \|\nu \cdot \boldsymbol{x}\|_2 \approx 1,$$

we may use $\nu$ in place of $\|\boldsymbol{x}\|_2$ for normalizing $\boldsymbol{x}$. If such normalizing factors $\nu$ can be found for all operation primitives, approximately normalized representations should be achievable without the need for exact normalization operations. It is clear that for such normalizing factors to exist, the norms of $\boldsymbol{x}$ have to concentrate closely around some value $\nu^{-1}$. Fortunately, under certain conditions, such behavior can indeed be observed.

**Theorem 1** (Concentration of Lipschitz functions on the sphere). *[12, pp. 106–109]*

*Let $\boldsymbol{x} \sim \mathcal{U}(S^{d-1})$ be a random vector uniformly distributed on the Euclidean unit sphere $S^{d-1} = \{\boldsymbol{x} \in \mathbb{R}^d : \|\boldsymbol{x}\|_2 = 1\}$ and let $f : S^{d-1} \to \mathbb{R}$ be a Lipschitz function. Then, for every $t \geq 0$,*

$$\mathbb{P}\{|f(\boldsymbol{x}) - \mathbb{E}[f(\boldsymbol{x})]| \geq t\} \leq 2 \exp\left(-\frac{cdt^2}{\|f\|_{Lip}^2}\right),$$

*where $c > 0$ and $\|f\|_{Lip}^2$ denotes the Lipschitz norm (smallest Lipschitz constant) of $f$.*

In words, Theorem 1 tells us that the deviation of $f(\boldsymbol{x})$ from its expected value decays exponentially in the dimension $d$, assuming $f$ is Lipschitz and $\boldsymbol{x} \sim \text{Unif}(S^{d-1})$. This implies that in high dimensions, $f(\boldsymbol{x})$ is likely to be close to $\mathbb{E}[f(\boldsymbol{x})]$, a phenomenon often referred to as *concentration of measure*.

In our setting, we consider functions of the form $f(\boldsymbol{x}) = \|g(\boldsymbol{x})\|_2$, where $g(\boldsymbol{x})$ denotes the output of a network component (e.g., a feedforward layer) given input $\boldsymbol{x}$. Assuming $g(\cdot)$ is Lipschitz and inputs $\boldsymbol{x}$ are normalized and approximately uniformly distributed on the sphere (e.g., by sampling from a Gaussian and renormalizing [12, p. 52]), the conditions of Theorem 1 are approximately satisfied. Consequently, the output norm $\|g(\boldsymbol{x})\|_2$ may concentrate around its expected value.

While these assumptions do not strictly hold during training, we may still observe concentration empirically, particularly due to the high dimensionality of representations. This "*blessing of dimensionality*" can justify approximating $\|g(\boldsymbol{x})\|_2$ by its expected value $\nu^{-1} := \mathbb{E}[\|g(\boldsymbol{x})\|_2]$ for the purpose of normalization.

## 4.2 Derivation of the normalizing factors $\nu$

Motivated by concentration effects in high dimensions, we derive normalizing factors for squared norms $\sqrt{\mathbb{E}[\|\boldsymbol{x}\|_2^2]}$, and then use $\nu^{-1} = \sqrt{\mathbb{E}[\|\boldsymbol{x}\|_2^2]}$. Computing $\nu$ this way should generally lead to an overestimation due to Jensen's inequality, as $\sqrt{\mathbb{E}[\|\boldsymbol{x}\|_2^2]} \geq \mathbb{E}[\sqrt{\|\boldsymbol{x}\|_2^2}] = \mathbb{E}[\|\boldsymbol{x}\|_2]$. However, for sufficiently high-dimensional $\boldsymbol{x}$, the bound tends to an equality due to the concentration effect. The derivation for the network components used for anTransformer is described below. A normalization of the attention matrix is described in Appendix D since it is not part of the architecture.

| | GPT+ | nGPT | anGPT (ours) |
|---|---|---|---|
| Embed | $\boldsymbol{h} \leftarrow W_e \boldsymbol{x_{\mathbf{in}}}$ | $\boldsymbol{h} \leftarrow W_e \boldsymbol{x_{\mathbf{in}}}$ | $\boldsymbol{h} \leftarrow W_e \boldsymbol{x_{\mathbf{in}}}$ |
| MHA | $\boldsymbol{h}_a \leftarrow \mathrm{rms}(\boldsymbol{h}) \cdot \boldsymbol{\gamma_a}$ 
 $\boldsymbol{q}, \boldsymbol{k}, \boldsymbol{v} \leftarrow W_{qkv} \boldsymbol{h_a}$ 
 $\boldsymbol{k} \leftarrow \mathrm{norm}(\boldsymbol{k})$ 
 $\boldsymbol{q} \leftarrow \mathrm{norm}(\boldsymbol{q})$ 
 $A \leftarrow \mathrm{softmax}(\boldsymbol{q}\boldsymbol{k}^T \cdot g)$ 
 $\boldsymbol{h}_a \leftarrow W_p(A\boldsymbol{v})$ | $\boldsymbol{q}, \boldsymbol{k}, \boldsymbol{v} \leftarrow W_{qkv} \boldsymbol{h}$ 
 $\boldsymbol{k} \leftarrow \mathrm{norm}(\boldsymbol{k}) \cdot \boldsymbol{s_k}$ 
 $\boldsymbol{q} \leftarrow \mathrm{norm}(\boldsymbol{q}) \cdot \boldsymbol{s_q}$ 
 $A \leftarrow \mathrm{softmax}(\boldsymbol{q}\boldsymbol{k}^T \cdot \sqrt{d_k})$ 
 $\boldsymbol{h}_a \leftarrow W_p(A\boldsymbol{v})$ 
 $\boldsymbol{h}_a \leftarrow \mathrm{norm}(\boldsymbol{h}_a)$ | $\boldsymbol{q}, \boldsymbol{k}, \boldsymbol{v} \leftarrow W_{qkv} \boldsymbol{h} \cdot \nu_{qkv}$ 
 $\boldsymbol{k} \leftarrow \mathrm{norm}(\boldsymbol{k})$ 
 $\boldsymbol{q} \leftarrow \mathrm{norm}(\boldsymbol{q})$ 
 $A \leftarrow \mathrm{softmax}(\boldsymbol{q}\boldsymbol{k}^T \cdot g)$ 
 $\boldsymbol{h}_a \leftarrow W_p(A\boldsymbol{v}) \cdot \nu_p$ 
 $\boldsymbol{h}_a \leftarrow \mathrm{norm}(\boldsymbol{h}_a)$ |
| Residual | $\boldsymbol{h} \leftarrow \boldsymbol{h} + \boldsymbol{h}_a$ | $\boldsymbol{h} \leftarrow \mathrm{norm}(\boldsymbol{h} + \boldsymbol{\alpha}_a(\boldsymbol{h}_a - \boldsymbol{h}))$ | $\boldsymbol{h} \leftarrow (\boldsymbol{h} + \boldsymbol{\alpha}_a(\boldsymbol{h}_a - \boldsymbol{h})) \cdot \nu(\boldsymbol{\alpha}_a)$ |
| MLP | $\boldsymbol{h}_m \leftarrow \mathrm{rms}(\boldsymbol{h}) \cdot \boldsymbol{\gamma_m}$ 
 $\boldsymbol{u}, \boldsymbol{z} \leftarrow W_{uz} \boldsymbol{h}$ 


 $\boldsymbol{h}_m \leftarrow \boldsymbol{u} \cdot \mathrm{SiLU}(\boldsymbol{z})$ 
 $\boldsymbol{h}_m \leftarrow W_d \boldsymbol{h}_m$ | $\boldsymbol{u}, \boldsymbol{z} \leftarrow W_{uz} \boldsymbol{h}$ 
 $\boldsymbol{u} \leftarrow \boldsymbol{u} \cdot \boldsymbol{s_u}$ 
 $\boldsymbol{z} \leftarrow \boldsymbol{z} \cdot \boldsymbol{s_z} \cdot \sqrt{d}$ 
 $\boldsymbol{h}_m \leftarrow \boldsymbol{u} \cdot \mathrm{SiLU}(\boldsymbol{z})$ 
 $\boldsymbol{h}_m \leftarrow W_d \boldsymbol{h}_m$ 
 $\boldsymbol{h}_m \leftarrow \mathrm{norm}(\boldsymbol{h}_m)$ | $\boldsymbol{u}, \boldsymbol{z} \leftarrow W_{uz} \boldsymbol{h} \cdot \nu_{uz}$ 


 $\boldsymbol{h}_m \leftarrow \boldsymbol{u} \cdot \mathrm{SiLU}(\boldsymbol{z}) \cdot \nu_{acf}$ 
 $\boldsymbol{h}_m \leftarrow W_d \boldsymbol{h}_m \cdot \nu_d$ 
 $\boldsymbol{h}_m \leftarrow \mathrm{norm}(\boldsymbol{h}_m)$ |
| Residual | $\boldsymbol{h} \leftarrow \boldsymbol{h} + \boldsymbol{h}_m$ | $\boldsymbol{h} \leftarrow \mathrm{norm}(\boldsymbol{h} + \boldsymbol{\alpha}_m(\boldsymbol{h}_m - \boldsymbol{h}))$ | $\boldsymbol{h} \leftarrow (\boldsymbol{h} + \boldsymbol{\alpha}_m(\boldsymbol{h}_m - \boldsymbol{h})) \cdot \nu(\boldsymbol{\alpha}_m)$ |
| Head | $\boldsymbol{h} \leftarrow \mathrm{rms}(\boldsymbol{h}) \cdot \boldsymbol{\gamma_h}$ 
 $\mathrm{logits} \leftarrow W_h \boldsymbol{h}$ | $\mathrm{logits} \leftarrow \boldsymbol{s_Z} \cdot (W_h \boldsymbol{h})$ | $\mathrm{logits} \leftarrow \boldsymbol{s_Z} \cdot (W_h \boldsymbol{h})$ |

Table 1: Comparison between GPT implementation with SwiGLU, RMSnorm, and QK-norm (GPT+), the normalized Transformer (nGPT), and our approximated normalized GPT (anGPT). We define $\mathrm{rms}(\mathbf{h}) = \mathbf{h}/\sqrt{1/N \sum_n^N h_n^2}$ and $\mathrm{norm}(\mathbf{h}) = \mathbf{h}/\|\mathbf{h}\|_2$ and colored learnable parameters green, constant scaling factors blue, and normalization factors purple.

**Linear map $\boldsymbol{Wx}$**    Assume $\boldsymbol{x} \in \mathbb{R}^d$ and $\boldsymbol{W} \in \mathbb{R}^{r \times d}$ with entries $x_i, w_{ij} \sim \mathcal{N}(0, 1)$ and normalized afterwards such that $\|\boldsymbol{x}\|_2 = 1$ and $\|\boldsymbol{w}_i\|_2 = 1$ where $\boldsymbol{w}_i$ denote the rows of $\boldsymbol{W}$. Then,

$$\mathbb{E}[\|\boldsymbol{W}\boldsymbol{x}\|_2^2] = \mathbb{E}\left[\sum_{i=1}^r (\boldsymbol{w}_i^\top \boldsymbol{x})^2\right] = \sum_{i=1}^r \mathbb{E}[(\boldsymbol{w}_i^\top \boldsymbol{x})^2] = r \cdot \frac{1}{d} = \frac{r}{d}$$

Note that specifically the assumption that $\boldsymbol{W}$ is normalized along its input dimension, i.e, the "weights of each neuron", has to be reflected in training in the form of constraints.

**Residual update**    Assume $\boldsymbol{x}, \boldsymbol{h} \in \mathbb{R}^d$ with independent entries $x_i, h_i \sim \mathcal{N}(0, 1)$ and normalized afterwards such that $\|\boldsymbol{x}\|_2 = 1$ and $\|\boldsymbol{h}\|_2 = 1$. For the classic residual update, this yields

$$\mathbb{E}[\|\boldsymbol{h} + \boldsymbol{x}\|_2^2] = \mathbb{E}[\boldsymbol{h}^\top \boldsymbol{h} + 2\boldsymbol{h}^\top \boldsymbol{x} + \boldsymbol{x}^\top \boldsymbol{x}] = 1 + 0 + 1 = 2.$$

Loshchilov et al. [5] proposed to replace the classic residual update by a linear interpolation (LERP) $\boldsymbol{h} \leftarrow \boldsymbol{h} + \boldsymbol{x}$ with $\boldsymbol{h} \leftarrow \boldsymbol{h} + \boldsymbol{\alpha}_a(\boldsymbol{x} - \boldsymbol{h})$ and $\alpha > 0$ which leads to the benefit of explicitly learning the impact of a layer.

$$\begin{aligned}
\mathbb{E}[\|\boldsymbol{h} + \alpha(\boldsymbol{x} - \boldsymbol{h})\|_2^2] &= \mathbb{E}[\|(1 - \alpha)\boldsymbol{h} + \alpha\boldsymbol{x}\|_2^2] \\
&= (1 - \alpha)^2 \mathbb{E}[\boldsymbol{h}^\top \boldsymbol{h}] + 2\alpha(1 - \alpha)\mathbb{E}[\boldsymbol{h}^\top \boldsymbol{x}] + \alpha^2 \mathbb{E}[\boldsymbol{x}^\top \boldsymbol{x}] \\
&= 1 - 2\alpha + \alpha^2 + 0 + \alpha^2 = 1 - 2\alpha + 2\alpha^2.
\end{aligned}$$

The same derivation also holds for element-wise multiplication with a vector $\boldsymbol{\alpha}$ instead of a scalar.

**Activation function**    Due to the nonlinearity of activation functions, we resort to numerical computation of the (squared) norm of the activations. For the calculations, we assume a uniform distribution on a sphere $S^{d-1}$. Specifically, inputs $x_i \sim \mathcal{N}(0, 1)$ and normalize afterwards such that $\|\boldsymbol{x}\|_2^2 = 1$. For computing the expected value, quadrature methods or Monte Carlo estimation can be used.

**Constraining Parameters**    As mentioned in the derivation of the linear map, we require our weights to be normalized along the input dimension. However, this does not allow to express zero vectors (the zero vector does not lie on the hypersphere). We thus only employ a bound $\|\boldsymbol{w}\|_2 \leq 1$ similar to [13], which allows the weights to adapt more freely. Consequently, our representations are not necessarily *normalized* but bounded, and tending towards the boundary, see Figure P.1. These representations

are therefore termed *compact*. Bounding the parameters also eliminates the need for regularization, such as weight decay. In line with Loshchilov et al. [5], we found in preliminary experiments that we do not require learning rate warm-up when initializing the parameters already normalized along the input dimension. This may be due to the more stable behavior of optimization, see e.g., Figure B.1 for the variance of the first moment in Adam.

### 4.3 Approximal normalized GPT (anGPT)

In the following, we describe our approximal normalized GPT model, *anGPT*, inspired by [5]. In contrast to nGPT, we perform normalization consistent with the assumption above, while nGPT normalizes along the residual dimension (to stay on a hypersphere). This leads to different normalization dimensions for linear maps in the input and output of an attention or feed-forward layer. Let $d_m$, $d_h$, $d_f$ denote the model, attention head, and up-scaled dimension, respectively. In the feed-forward MLP, $l$ as the number of layers, and $d_v$ as the vocabulary size. An overview of our architectural modifications, including nGPT, can be found in Table 1.

**Replace norm and residual update**   First of all, we remove all classical normalization layers and replace them with a post-L2-normalization $\text{norm}(x) = x/\|x\|_2$. Removing normalization completely leads to unstable training. The architecture replaces the classic residual update with LERP and replaces the learnable per-element affine parameter from the pre-norm layer with a learnable interpolation parameter $\boldsymbol{\alpha}$.

**Add normalization factors**   We add constant normalization factors in the attention layer for the query, key, and value map, but due to the reshape into the head dimension, we consider the head dimension as the output dimension $\nu_{qkv} = \sqrt{d_m/d_h}$. Similarly, we add a normalization factor for the output map $\nu_p = \sqrt{d_h/d_m}$ but with $d_h$ as input dimension. In the feed forward layer we add three normalization factors $\nu_{uz} = \sqrt{d_m/(4d_m)}$, $\nu_d = \sqrt{(4d_m)/d_m}$, and $\approx 3.74$ (estimated via Monte Carlo) for upscaling, downscaling and the activation function, respectively. The normalization factor for the residual LERP update is a function of $\boldsymbol{\alpha}$ and calculated at each step through $\nu(\boldsymbol{\alpha}) = 1 - 2\boldsymbol{\alpha} + 2\boldsymbol{\alpha}^2$. A list of normalization factors for each primitive can be found in Appendix E. During inference, we can subsume constant normalization factors and the logits scaling into the model parameters. Since we need no more normalization layers than the GPT model, we expect the same inference time.

**Logits scaling**   anGPT removes the RMSnorm before the head linear since the representation is already normalized. However, to scale the logits, a learnable scaling vector $\boldsymbol{s_z} \in \mathbb{R}^{d_v}$ is added, similar to nGPT. Scaling before the head linear was also tested but reduced performance.

**Parameter Reparameterization for Uniform Optimization**   Following nGPT [5], we employ a reparameterization scheme to ensure uniform optimization dynamics across parameter types. For any trainable scaling parameter $s_a$ (e.g., $\alpha_A$, $\alpha_M$, $s_z$), we optimize a surrogate parameter $\hat{s}_a$ and compute:

$$s_a = \frac{s_{a,\text{init}}}{s_{a,\text{scale}}} \cdot \hat{s}_a$$

The surrogate is initialized as $\hat{s}_a^{(0)} = s_{a,\text{scale}}$, ensuring $s_a^{(0)} = s_{a,\text{init}}$. This reparameterization ensures all stored parameters have comparable magnitude $\sim s_{a,\text{scale}}$, enabling Adam's adaptive learning rate mechanism to work uniformly across the network. The effective learning rate for updates to $s_a$ becomes $\alpha_{\text{eff}} = \alpha \left(s_{a,\text{init}}/s_{a,\text{scale}}\right)^2$.

## 5   Experiments

In the following, we describe multiple LLM pretraining experiments comparing anGPT to GPT and nGPT. The experiments use SlimPajama [14] with ~$627B$ tokens and train for $< 1$ epoch. All experiments employ the GPT-NeoX tokenizer with a 50k vocabulary size [15] and a context window of 2048 tokens. As a baseline, we extend the vanilla GPT architecture with a SwiGLU activation function [16], rotary position embedding [17], RMS normalization [8], and QK normalization [18] and dub the resulting model *GPT+*. All models are trained with Adam [11] and, in the case of GPT+,

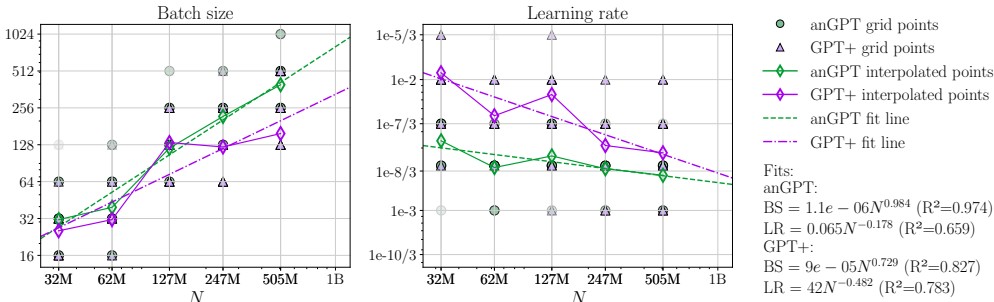

Figure 2: Scaling trend fits for optimal batch size and learning rate as functions of model size $N$. Grid point markers are shaded by excess loss relative to all configurations for this parameter. Diamond markers show the two-stage interpolation-based estimates of optimal hyperparameters. Dashed lines represent fitted power laws using the estimated optimal hyperparameters.

with AdamW [19]. For learning rate scheduling, a cosine learning rate annealing [20] is employed. If learning rate warm-up is used, $10\%$ of the total training steps are dedicated to warm-up. For adapting the effective learning rate of scaling parameters, we set $s_{a,\text{init}} = 0.01$ for anGPT and $s_{a,\text{init}} = 1/\sqrt{(d)}$ for nGPT. The following experiments compare anGPT to nGPT and GPT+. Hyperparameters are reported in Appendix F and comparisons with related work in Appendix A.

## 5.1 Hyperparameter Scaling Trends

To determine optimal learning rates and batch sizes, hyperparameter scaling trends were derived across model sizes from $N = 32M$ to $N = 0.5B$ parameters (Configuration in Table F.1). The methodology follows Porian et al. [21] and DeepSeek-AI et al. [10]. We performed a grid search for each model size over the batch size and learning rate with a Chinchilla optimal token budget ($D = 20N$, [22]). Optimal batch size and learning rate for each model size were obtained via a two-stage interpolation process adapted from Porian et al. [21]. The estimates for the optimal values are shown in Figure 2. The process is described in detail in Appendix G with full hyperparameter sweep results in Figure G.1. When fitting hyperparameter scaling trends for the optimal parameter selection, we found that anGPT can use larger batch sizes for larger model sizes, which could be beneficial for scaling the pretraining to a large number of workers for achieving speedup. For the scaling behavior of the learning rate, we found a small negative exponent for anGPT, so the learning rate decreases gradually as models get larger, which is expected behavior in line with previous works [23, 10].

## 5.2 Performance comparison

To compare our approach to GPT+ and nGPT, we trained a $0.5B$ model of each architecture on token budgets from $5B$ to $70B$ ($0.5\times$ to $7\times$ Chinchilla optimal). For GPT+ and anGPT, we used the optimal hyperparameters from the previous grid search and performed an additional grid search for nGPT. Figure 3 reports the results and convergence speedup against GPT+. Results show that anGPT and nGPT outperform GPT+ across all token budgets, with nGPT performing better at smaller token budgets and reaching comparable performance at larger ones. Convergence measurements reveal an average speedup factor of $1.4\times$ for anGPT and $1.29\times$ for nGPT compared to GPT+. Similar to Loshchilov et al. [5], convergence speedup measurements against GPT+ without QK normalization yield speedup factors of $2.0\times$ for anGPT and $1.8\times$ for nGPT, see Figure H.2. We discuss the discrepancy to [5] reported speedups in Appendix H. In addition, we performed performance comparisons on different token budgets for the $250M$ and $1.0B$ model sizes and find the  40% convergence speedup observed for the 0.5B model consistent. Convergence plots for both models are provided in Appendix I.

Table 2 reports the average runtime per training step for all three architectures. anGPT shows an increase of ~3% and nGPT shows ~9% increase. The additional runtime is attributed to the additional scaling factors, the norm implementation, and the additional norm for nGPT. During inference, similar

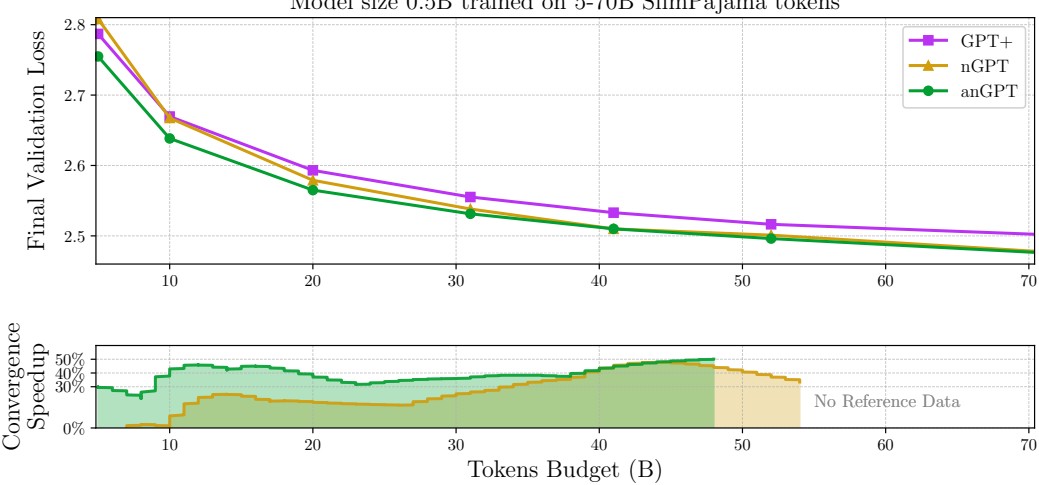

Figure 3: Training the $0.5B$ model up to $7\times$ Chinchilla optimal token budget. Each point is the final validation loss of a full training run with the training budget noted on the abscissa. Below, we measure the convergence speed-up against the GPT+ model with QK normalization.

runtimes were observed for GPT+ and anGPT but a ~3% increase for nGPT, due to the additional normalization operations.

## 5.3 Comparing performance via compute-dependent scaling law

We further compare GPT+ and anGPT by training model sizes from $32M$ to $1B$ parameters on token budgets from $0.5\times$ to $5\times$ Chinchilla optimal. This enables investigation of the scaling behavior of the new anGPT architecture. Figure 4 shows the results and scaling fits. Given the same compute budget, anGPT outperforms GPT+ on any model size and compute budget. These results were used to derive scaling laws using the approach of Hoffmann et al. [22], as described in Appendix J. Both GPT+ and anGPT exhibit nearly identical estimated scaling law exponents, implying that the improvement of validation loss is not significantly different across these architectures.

Table 2: Runtime Comparison with a $0.5B$ parameter model on a GPU node with 4 A100 (40GB) GPUs with a sequence length of 2048 and a batch size of 8. The experiments use torch.compile with default settings.

| Model | Avg. Runtime per Step | Rel. Increase (%) |
|---|---|---|
| GPT+ | 0.1416 | - |
| anGPT | 0.1455 | 2.75 |
| nGPT | 0.1552 | 9.60 |

## 5.4 Downstream Evaluation

To verify that pretraining improvements transfer beyond perplexity metrics, we evaluate our models on standard benchmarks. The 1B anGPT model consistently outperforms GPT+ across six benchmarks and three training budgets, with improvements ranging from 3% to 22% depending on the task. Detailed downstream evaluation results are presented in Appendix K.

## 5.5 Ablation studies

To investigate the modifications added to a vanilla GPT architecture, small experiments were conducted on a $0.5B$ GPT model trained on $10B$ tokens from OpenWebText [24]. When adding QK norm to the baseline, it leads to a performance gain of $2.1\%$ as visualized in Figure 5. Additional benefits emerge from using nGPT, with further improvements from using weight bounds instead of weight normalizations. The replacement of the normalization layer after the LERP update by a normalization factor does not reduce the performance. For anGPT (additional normalization factors, different normalization dimension, removing scaling vectors), we see an additional gain of $0.4\%$. Ex-

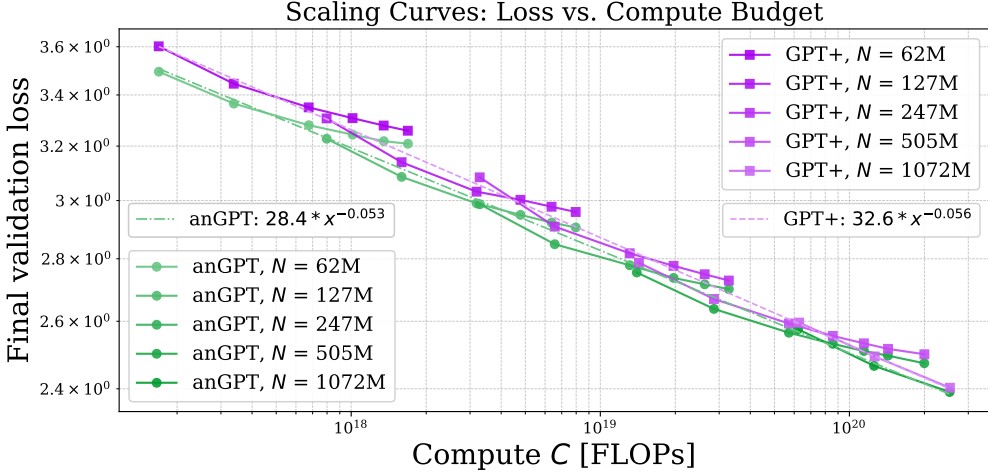

Figure 4: Training different model sizes on different token budgets. Each point represents a full training with the training budget noted on the abscissa. The scaling law is fitted for both architectures as described in Appendix J and indicated by the dashed line.

tensive ablation studies further assess the sensitivity of anGPT to variations in normalization factors. Our analysis shows that while the method is robust to small estimation errors (below 2%), the residual normalization factors are critical for training stability. Complete ablation results, including sensitivity to factor scaling and comparisons with different configurations, are provided in Appendix L.

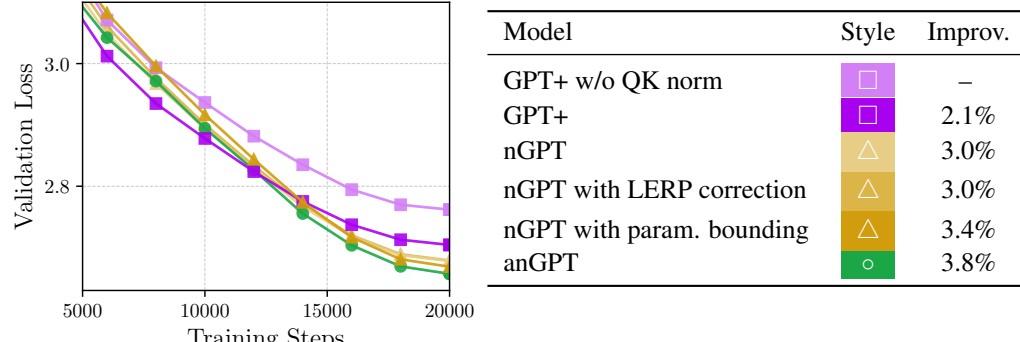

| Model | Style | Improv. |
|---|---|---|
| GPT+ w/o QK norm | ☐ | – |
| GPT+ | ☐ | 2.1% |
| nGPT | △ | 3.0% |
| nGPT with LERP correction | △ | 3.0% |
| nGPT with param. bounding | △ | 3.4% |
| anGPT | ○ | 3.8% |

Figure 5: We run ablation experiments with a $0.5B$ parameter model using $10B$ tokens from OpenWebText. Adding QK norm shows a performance gain. We modify nGPT by replacing the normalization of the LERP update with a normalization factor and, in addition, by bounding weights instead of normalizing them. The anGPT mainly replaces scaling vectors by normalization factors.

## 5.6 Analysis of anGPT

First of all, we analyze the learnable interpolation parameters $\alpha$ and find adaptive feature utilization throughout training, with values increasing from 0.05 to 0.12–0.25. Details are provided in Appendix M. Further, we empirically verify that approximate normalization maintains stable norms throughout the network. Analysis shows that anGPT maintains near-unity norm ratios (0.86–1.86) across layers while GPT+ exhibits significant norm growth (up to $5.83\times$ in deeper layers). Detailed measurements are provided in Appendix N. Finally, we evaluate robustness to distribution shifts using pathological inputs. anGPT maintains better norm stability ($1.5\times$ change) compared to GPT+ ($3.4\times$ change), with no catastrophic failures. See Appendix O for detailed analysis.

# 6 Limitations

Despite the extensive experimental evaluation, we do not perform experiments with more than $7\times$ Chinchilla optimal token budgets; our approach could perform worse than nGPT or GPT+ training with larger budgets. We also did not perform multiple experiments with different random seeds to generate error bars due to the high cost of GPT pretraining. We do not evaluate on the downstream task and assume that the validation loss correlates with the downstream performance. It is also hard to get a meaningful downstream signal from small and short-trained LLMs. Lastly, we perform the majority of the experiments on only one dataset and only with the GPT architecture; the performance could diverge with different datasets and data modalities.

# 7 Discussion

In our preliminaries, we hypothesized that the benefits of normalization stem from two effects: stabilizing input scales across layers and preventing representation norm escalation. Our experimental results provide strong evidence supporting both hypotheses. First, we hypothesized that normalization enhances optimization, as confirmed by Figure B.1. This shows that anGPT exhibits significantly reduced variance in Adam's first moments compared to GPT+, without requiring learning rate warm-up. Second, our hypothesis addressed the "Curse of Depth." As seen in Figure 1, GPT+ shows input norms growing exponentially with depth, while anGPT successfully constrains this growth. Traditional normalization layers employ learnable parameters $\gamma$ that serve two purposes: stabilizing network activations while simultaneously participating in loss minimization. This conflation of roles complicates optimization dynamics. In anGPT, we deliberately decouple these concerns—normalization factors $\nu$ handle stabilization exclusively, while the remaining parameters focus solely on minimizing the loss. This decoupling eliminates the need for warm-up while preserving favorable scaling properties. Since weight decay is also unnecessary, the practical impact is substantial: hyperparameter tuning becomes simpler and scaling law derivation more efficient, as fewer parameters need to be tuned.

Interestingly, the theoretical assumptions motivating our normalization factors do not strictly hold during training and are even violated by design. This occurs because weights are bounded rather than normalized, leading to compact rather than perfectly normalized representations. Nevertheless, we still observe concentration empirically (see Figure P.1), particularly due to the high dimensionality of representations. This robustness to assumption violations actually strengthens our approach, demonstrating that exact normalization is unnecessary to achieve the benefits traditionally associated with normalization layers. The key insight is that decoupling stabilization from loss minimization enables $1.4\times$ faster convergence compared to GPT with QK norm, while adding only minor computational overhead (see Figure 3).

Looking forward, several directions warrant further investigation. The compact representation space maintained by anGPT makes it particularly amenable to reduced-precision training. The bounded nature of activations and weights could enable efficient FP8 training without the numerical instabilities typically associated with low-precision arithmetic in unbounded architectures. Additionally, extending the approximate normalization framework to other architectures such as vision transformers and diffusion models could yield similar efficiency gains. Another promising avenue is exploring whether alternative optimizers might be more suitable for normalized architectures than Adam, as the bounded parameter space and stable gradient flow could benefit from optimization algorithms specifically designed for compact spaces.

# 8 Conclusion

We presented anGPT, an approximately normalized transformer that achieves faster convergence through scalar multiplication-based normalization. By leveraging the concentration of norms in high-dimensional spaces and decoupling stabilization from loss minimization, our method eliminates the need for weight decay and learning rate warm-up while maintaining only 3% computational overhead. Overall, anTransformer demonstrates that the benefits of consistent normalization, such as convergence speedup and fewer hyperparameters, can be achieved with minimal computational overhead, and provides a promising approach to training large language models with predictable scaling behavior.

## Acknowledgements

This research was funded by the Deutsche Forschungsgemeinschaft (DFG, German Research Foundation) under grant number 417962828. We acknowledge funding by the European Union (via ERC Consolidator Grant DeepLearning 2.0, grant no. 101045765). Views and opinions expressed are, however, those of the author(s) only and do not necessarily reflect those of the European Union or the European Research Council. Neither the European Union nor the granting authority can be held responsible for them.



Further, we acknowledge funding by the Federal Ministry of Education and Research of Germany (BMBF) under grant no. 01IS22094B (WestAI - AI Service Center West), under grant no. 01IS24085C (OPENHAFM) and under the grant 16HPC117K (MINERVA), as well as co-funding by EU from EuroHPC Joint Undertaking program under grant no. 101182737 (MINERVA) and from the Digital Europe Programme under grant no. 101195233 (openEuroLLM).

We gratefully acknowledge the Gauss Centre for Supercomputing e.V. for funding this work by providing computing time through the John von Neumann Institute for Computing (NIC) on the supercomputer JUWELS Booster at Jülich Supercomputing Centre (JSC), EuroHPC Joint Undertaking for awarding this project access to the EuroHPC supercomputer LEONARDO, hosted by CINECA (Italy) and the LEONARDO consortium through an EuroHPC Extreme Access grant EHPC-EXT-2023E02-068, storage resources on JUST granted and operated by JSC and supported by Helmholtz Data Federation (HDF), compute resources provided via WestAI compute grant "Measuring and enhancing advanced reasoning capabilities of foundation models via local model deployment" (westai0007, westai0066) at JSC and compute resources provided via JSC at JURECA.

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

# Appendix

# A    Related Work

Previous approaches have addressed the "Curse of Depth" [6] through depth-specific scaling [25], initialization strategies [26], or hybrid normalization schemes [27]. Sun et al. [6] introduced a constant scaling factor depending on the layer number after the pre-normalization layer. This reduces the effect of increasing variance on the residual, but in contrast to our approach, it does not aim to eliminate the effect. In this work, we propose a comprehensive solution that normalizes the entire representation at each layer, ensuring all network components contribute effectively regardless of depth.

Franke et al. [13] introduces *Constrained Parameter Regularization* which bounds a statistical measure, like the $L_2$-norm, of learnable parameters using an augmented Lagrangian optimization instead of applying weight decay. Therefore, they introduce multiple initialization methods to find the right norm value. In contrast, our work bound all parameter matrices to one, and we also approximately normalized the representation space.

Multiple works proposed methods or architecture changes to remove or replace the normalization [28–31]. Most recently, Zhu et al. [32] proposed to replace the normalization layer with a *Dynamic Tanh* layer (DyT) based on the observation that normalization produces S-shaped input-output mappings. DyT consists of a tanh function with an input scaling scalar and output scaling vector. In contrast to our approach, DyT does not change the representation or weight space and only aims for an improved runtime.

**Comparison to related work**

We compare our anGPT approach to the constant scaling factor [6] and the dynamic tanh (DyT) replacement of the layer normalization [32]. We train both on a $0.5B$ GPT setting on $10B$ SlimPajama tokens (Chinchilla optimal) using the same configuration as for GPT+. However, we tune the learning rate for both approaches. We apply the additional learnable scaling factor for DyT as proposed and a $\alpha_0$ initialization of $1.0$ according to Table 12 in [32]. We include in the comparison nGPT and GPT+ and show the results in Figure A.1.

We see a strong performance drop using DyT. When using LN scaling, we find the results on par with GPT+ without scaling. nGPt outperforms GPT+ slightly, and anGPT shows the best performance in this setting.

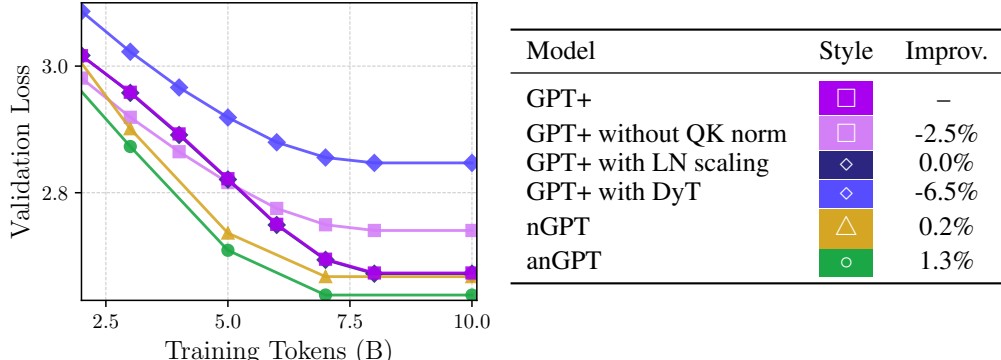

| Model | Style | Improv. |
|---|---|---|
| GPT+ | ◻ | – |
| GPT+ without QK norm | ◻ | -2.5% |
| GPT+ with LN scaling | ◇ | 0.0% |
| GPT+ with DyT | ◇ | -6.5% |
| nGPT | △ | 0.2% |
| anGPT | ○ | 1.3% |

Figure A.1: Comparison of GPT+, nGPT, and anGPT to the LN scaling [6] and DyT normalization [32] in a 0.5B GPT training on 10B SlimPajama token budget. We use the same configuration as for GPT+ and tune the learning rate.

# B Variance on the Adam momentum

To understand the effects of our normalization on training with Adam, we analyze the (relative) variance of the momentum vector $m$. The relative variance was computed by dividing each step of the timeseries $\mathbb{V}(m_t)$ by $\sum_{t=1}^{T} \mathbb{V}(m_t)$

We can see that for GPT+, warm-up allows the (relative) variances of the momentum terms to become small before the main training starts at 2000 steps, see Figure B.1. Without warm-up, see Figure B.2, we observe stronger peaks in the relative variance of the momentum vector, which likely destabilizes training. It can also be seen that the (relative) variance of the momentum vector for anGPT starts mostly at zero for all parameters and develops gradually with little deviation between parameters.

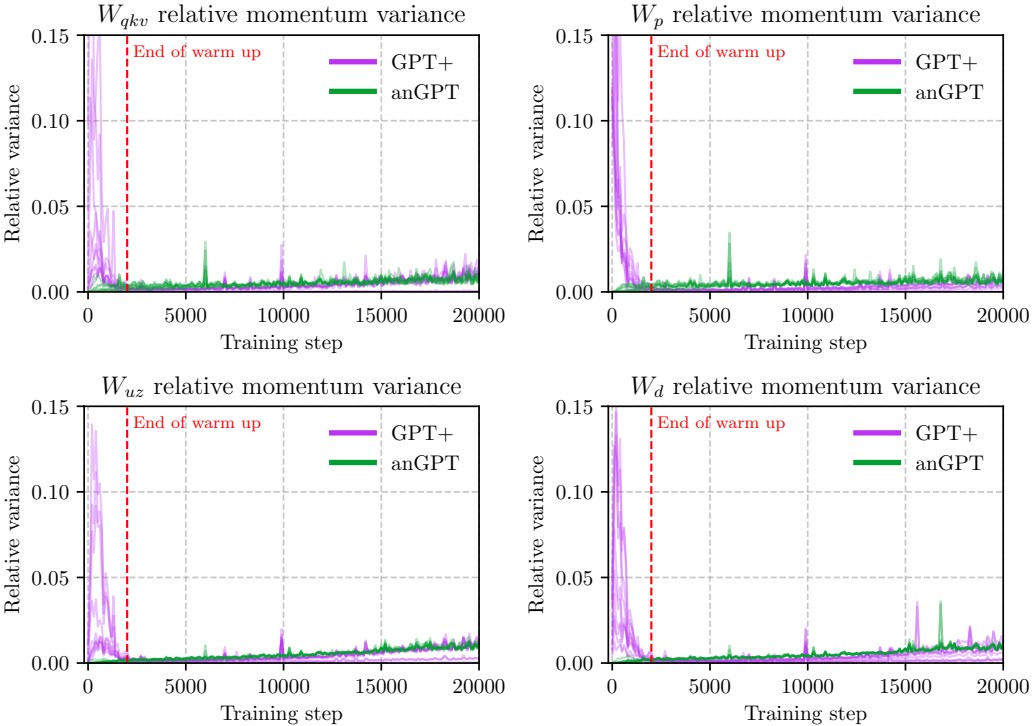

Figure B.1: The Variance of the Adam momentum for the Transformer parameter groups during training. The traces correspond to individual layers. Each subplot shows the per-step variance of Adam's first moment estimates for one weight matrix, relative to each layer's total variance over all training steps. Despite learning-rate warm-up, GPT+ shows a variance spike in all four parameter groups at the start of the training.

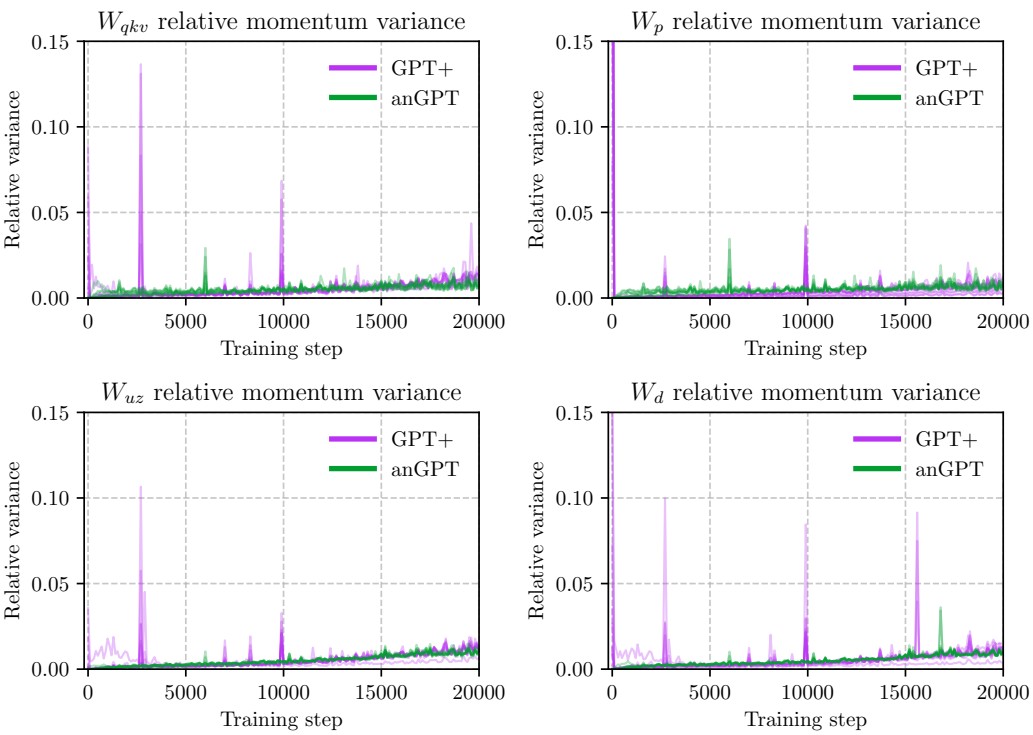

Figure B.2: The Variance of the Adam momentum for the Transformer parameter groups during training. The traces correspond to individual layers. Each subplot shows the per-step variance of Adam's first moment estimates for one weight matrix, relative to each layer's total variance over all training steps. GPT+ is trained without a learning rate warm-up.

## C  RMSNorm $\gamma$-norms and block-input norms in GPT+

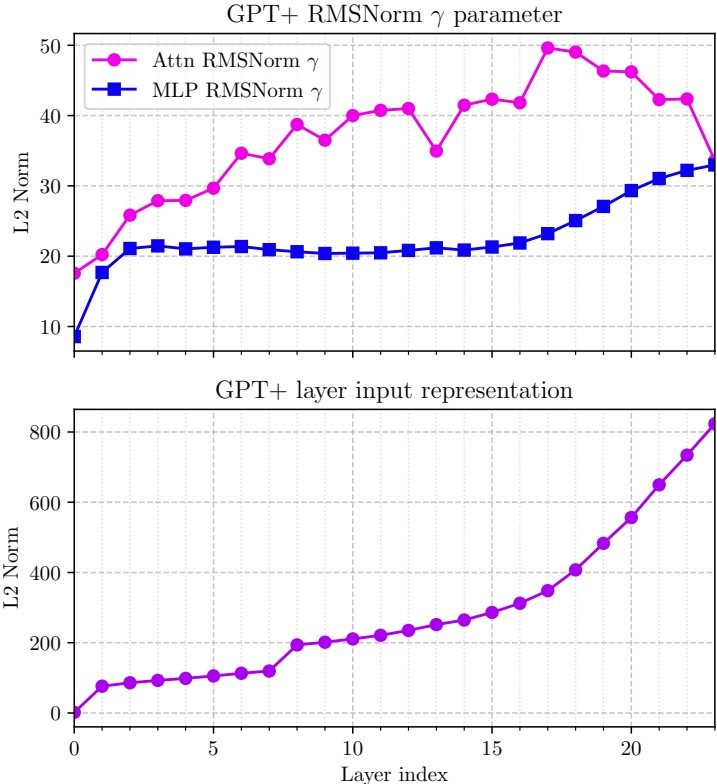

Figure C.1: The RMSNorm $\gamma$ norm and layer input norm for each layer in GPT+ after the final training. We see the growing norm on the residual and a slightly correlated growth of the norm in the $\gamma$ parameter.

## D Normalizing the Attention Matrix

To apply the normalization for the linear map to $\boldsymbol{Av}$, we require the rows $\boldsymbol{a}_i$ of $\boldsymbol{A}$ to be normalized. Assuming the entries $a_{ij}$ of $\boldsymbol{A} \in \mathbb{R}^{s \times s}$ are identically distributed, and the softmax computation is applied along the rows $\boldsymbol{A}$, we know that $\sum_j a_{ij} = 1$. By symmetry (identical distribution), we know that $\mathbb{E}[a_{ij}] = 1/s$. For the rows $\boldsymbol{a}_i$ to be normalized, it is required that $\|\boldsymbol{a}_i\|_2^2 = \sum_j a_{ij}^2 = 1$, i.e., $\mathbb{E}[a_{ij}] = 1/\sqrt{s}$. We therefore scale the attention matrix by $s/\sqrt{s} = \sqrt{s}$. If a mask for causal attention is used, we know the number of zero entries in each row. Thus, the expected value of the nonzero entries in the $r$-th row is $\mathbb{E}[a_{rj}] = 1/r$, so the $r$-th row is scaled by $r/\sqrt{r} = \sqrt{r}$.

Note that the normalization factors are likely less useful for attention as the expected values effectively model dense attention scores, while sparse attention is usually more realistic. Hence, calculating useful normalization factors for the attention matrix likely requires information about the number of effectively nonzero attention scores. Unfortunately, this information is often not directly accessible due to the use of FlashAttention [33]. Approaches to leverage this information for effective normalization may be seen as directions for future work.

## E Explicit Normalization Factors

We derive the normalization factors using $\nu_g = 1/\sqrt{\mathbb{E}[\|g(x)\|_2^2]}$ for linear maps and Monte Carlo estimation for activation functions. The specific factors are:

- Query, Key, Value projections: $\nu_{qkv} = \sqrt{d/h}$ for $W_{qkv} \in \mathbb{R}^{h \times d}$
- Output projection: $\nu_p = 1$ for $W_p \in \mathbb{R}^{d \times d}$
- Up projection: $\nu_{uz} = \sqrt{d/f}$ for $W_{uz} \in \mathbb{R}^{f \times d}$
- Activation function: $\nu_{acf} = 3.74$ (via Monte Carlo with $10^5$ samples)
- Down projection: $\nu_d = \sqrt{f/d}$ for $W_d \in \mathbb{R}^{d \times f}$
- Residual interpolation: $\nu(\alpha) = 1/\sqrt{\alpha^2 + (1 - \alpha)^2}$

where $d$ is the model dimension, $h$ is the head dimension (64 in our experiments), and $f$ is the feed-forward dimension (typically $4d$).

# F   Training Details

In our experiments, we use model sizes from $32M$ up to $1B$ parameters and list the architecture hyperparameters in Table F.1. For the optimization, we use Adam and AdamW and list the corresponding training hyperparameters in Table F.2. We used two datasets in this paper, SlimPajama (Apache 2.0 license) [14] and OpenWebText (Creative Commons Zero v1.0) [24][1]. SlimPajama provides a validation set, and for OpenWebText, we used 10k randomly selected documents as a validation set.

We implemented our experiments in PyTorch 2.6 [34] and used Flash Attention 0.7.3 [33]. All plots are generated with Matplotlib [35]. We performed all experiments on a research cluster with $4\times$ A100 40GB GPU nodes and used in total about 30k GPU hours. The smallest experiments are around 1 GPU hour, and the largest are up to 750 GPU hours. An open-source implementation of anGPT is available at `https://github.com/automl/anGPT`.

Table F.1: The different GPT-style language model architectures used in this work and the total parameter counts. All models use a vocabulary size of 50,304 tokens. The difference between GPT+ and anGPT accrues due to the difference in head scaling size (vocabulary size instead of model dimension). The difference between GPT+ and nGPT comes from additional scaling vectors in each MLP ($2 \times d_{MLP}$) and MHA ($2 \times d_{model}$) layer.

| Model | 32M | 62M | 125M | 250M | 0.5B | 1B |
|---|---|---|---|---|---|---|
| Model Dimension ($d_{model}$) | 256 | 384 | 512 | 768 | 1024 | 1280 |
| Number of Layers ($n_{layers}$) | 6 | 10 | 18 | 18 | 24 | 36 |
| Number of Attn. Heads ($n_{heads}$) | 4 | 6 | 8 | 12 | 16 | 20 |
| Head Dim. ($d_k = d_{model}/n_{heads}$) | 64 | 64 | 64 | 64 | 64 | 64 |
| MLP Dim. ($d_{MLP} = 4 \times d_{model}$) | 1024 | 1536 | 2048 | 3072 | 4096 | 5120 |
| Parameters in GPT+ | 32.05M | 62.24M | 127.03M | 247.17M | 505.73M | 1.073B |
| Parameters in nGPT | 32.11M | 62.33M | 127.16M | 247.34M | 506.00M | 1.073B |
| Parameters in anGPT | 32.10M | 62.28M | 127.08M | 247.21M | 505.78M | 1.073B |

Table F.2: If not other specified, we used the following hyperparameters of the GPT+, nGPT, and anGPT training runs in the experiment section.

| Parameter | GPT+ | nGPT/anGPT |
|---|---|---|
| Gradient Clip Val | 1.0 | |
| Precision | bf16-mixed | |
| Optimizer | AdamW | Adam |
| Beta1 | 0.9 | |
| Beta2 | 0.95 | |
| Eps | $1.0 \times 10^{-9}$ | |
| Weigth decay | 0.1 | 0 |
| Lr Num warm-up Steps | 20% | 0 |
| Lr Decay Factor | 0.01 | |
| Lr Schedule | Cosine | |
| Param. Scale Init | - | $1/\sqrt{d_m}$ / 0.001 |
| Dropout | 0 | |
| Rotary Pos Embed | True | |
| Rotary Emb Fraction | 0.5 | |
| Use Bias | False | |
| Flash Attention | True | |
| Torch Compile | True | |
| Context size | 2048 | |

---

[1]Both datasets are accessible on Huggingface: `https://huggingface.co/datasets/cerebras/SlimPajama-627B` and `https://huggingface.co/datasets/Skylion007/openwebtext`

# G   Fitting hyperparameter scaling trends

To investigate the optimal hyperparameters and scaling trends of the new anGPT architecture, we performed a grid search on different scales and used the extrapolated optimal configuration to train multiple models on different token budgets. We orient our procedure on Porian et al. [21] work investigating the discrepancies in compute-optimal scaling of language models between Kaplan et al. [36] and Hoffmann et al. [22].

For the grid search, we performed training runs with GPT+ and anGPT on the SlimPajama dataset with different model scales from $32M$ to $0.5B$ parameters. We train a *Chinchilla optimal* token budget of $20\times$ the number of training parameters [36]. Similar to Porian et al. [21], we used an Adam $\beta_2$ parameter of $0.99$ for experiments below $100M$ parameters and $0.95$ above. We performed at least three experiments per scale and batch size with different learning rates, so that the best configuration is always in the middle of the parameter grid. Our raw results of the hyperparameter sweeps can be found in Figure G.1.

**Estimating the optimal batch size and learning rate via interpolation**

Our interpolation method for estimating the optimal batch size and learning rate closely follows the two-stage procedure proposed by Porian et al. [21]. In their method in the first stage, for each model size and batch size, the optimal learning rate was identified by performing Akima interpolation (in log-space) on the loss as a function of the learning rate, taking the lowest loss among three tested values of the hyperparameter $\beta_2$, and subsequently identifying the minimizing argument. In the second stage, they applied interpolation again, this time over batch sizes, using the previously interpolated minimal losses to pinpoint an optimal batch size. The final optimal learning rate for the identified batch size was obtained by interpolating the sequence of (batch size, minimizing learning rate) pairs and evaluating this interpolant at the determined optimal batch size.

Our approach mirrors this two-stage interpolation methodology but differs by utilizing pre-selected and fixed $\beta_2$ values, thus avoiding the need for optimization over $\beta_2$. As discussed in the previous section, we use an Adam $\beta_2$ parameter of $0.99$ for experiments below $100M$ parameters and $0.95$ above.

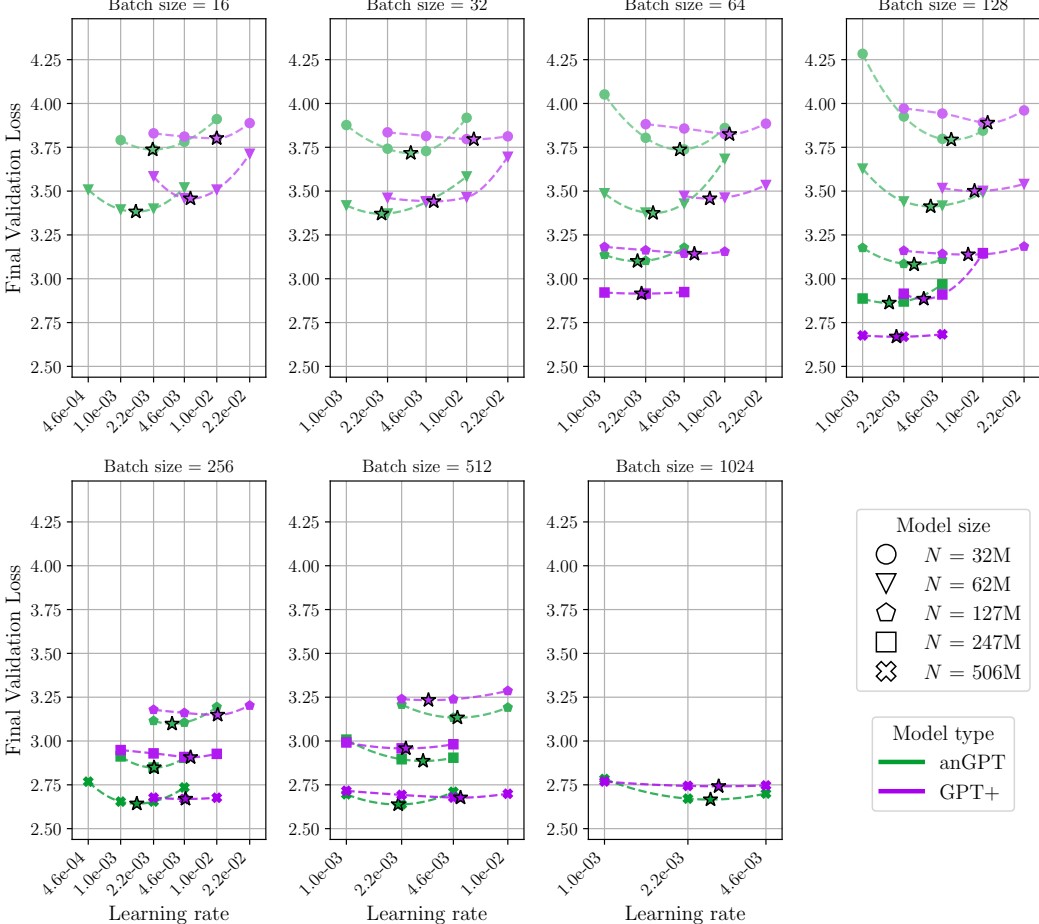

Figure G.1: Hyperparameter sweep results: The validation loss after $20 \times N$ training steps as a function of the learning rate for different model sizes $N$ and batch sizes. Plot design inspired by Wortsman et al. [37]. The stars with a black outline indicate the interpolated minimum learning rate as used in the first stage of the estimation of the optimal batch size and learning rate.

## H Comparison to nGPT

Since we compare our work with nGPT [5], we performed a sanity check with our code base and compared the results of the official nGPT implementation [38]. We reconstructed the values of the reported model performance across different learning rates from the repository in Figure H.1 and added our results. We find the final validation loss values for nGPT and GPT without QK norm match the reported numbers. We also found anGPT has a slightly lower final validation loss $(0.6\%)$ than nGPT, but, maybe more importantly, we also found that GPT with QK norm increases the GPT baseline performance substantially $(2.6\%$ lower final validation loss).

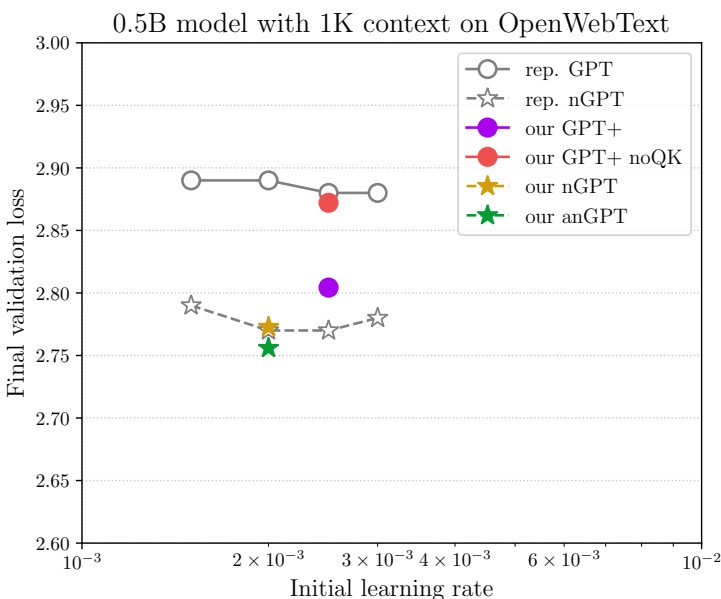

Figure H.1: We reconstructed the Figure from the nGPT Github repository and compared the results of our implementation with the official reported loss values (gray) from experiments with a 0.5B parameter model, 512 batch size, 1k context size, and on 5B OpenWebText tokens / 10k training steps.

This finding is in line with our experiment on SlimPajama, shown in Figure H.2. We find a higher convergence speedup when compared to GPT+ without QK norm. However, there is still a gap to the reported speedup factors of $4\times$ up to $20\times$ by Loshchilov et al. [5]. We hypothesize this could be due to different training data, training budget, hyperparameters, and/or the codebase.

In contrast to nGPT, we use a larger training corpus to perform LLM pretraining experiments without training multiple epochs. While Loshchilov et al. [5] used OpenWebText [24] with ~$9B$ tokens (32k tokenizer) and trained for multiple epochs (up to 50 epochs), we use SlimPajama [14] with ~$627B$ tokens (50k tokenizer) and train for $< 1$ epoch. Also, we trained only up to $\times 7$ Chinchilla optimal [22] token budgets while nGPT was trained on up to $\times 20$ Chinchilla optimal token budgets. Furthermore, we tuned the batch size and found a smaller batch size for GPT+ slightly better than for anGPT. We trained only on a context length of 2k tokens (since the average document length in SlimPajama is only 1k tokens) while nGPT was trained on up to 8k tokens context size. Lastly, the publicly available codebase on GitHub is different from the one used in the paper, as the author explains in GitHub Issue 6 [38]. Nevertheless, the nGPT codebase was very helpful for our work.

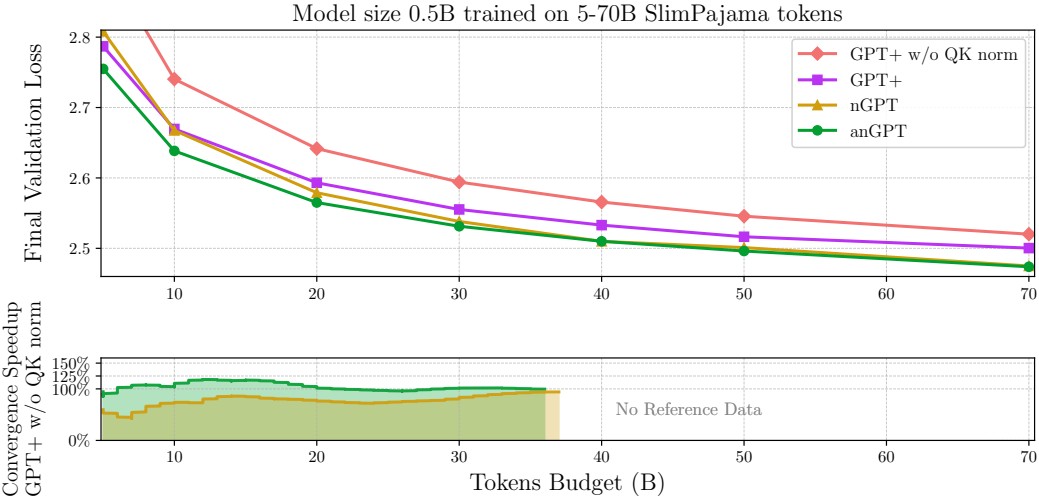

Figure H.2: Training the 0.5B model up to $7\times$ Chinchilla optimal token budget. Each point is a full training with the training budget noted on the abscissa. The plot below shows the convergence speed up against the GPT+ model *without QK normalization*.

# I  Convergence Analysis Across Model Scales

Figure I.1 shows convergence comparisons for 250M and 1B parameter models, demonstrating that the  40% speedup observed for 0.5B models (Figure 3 in main paper) is consistent across scales.

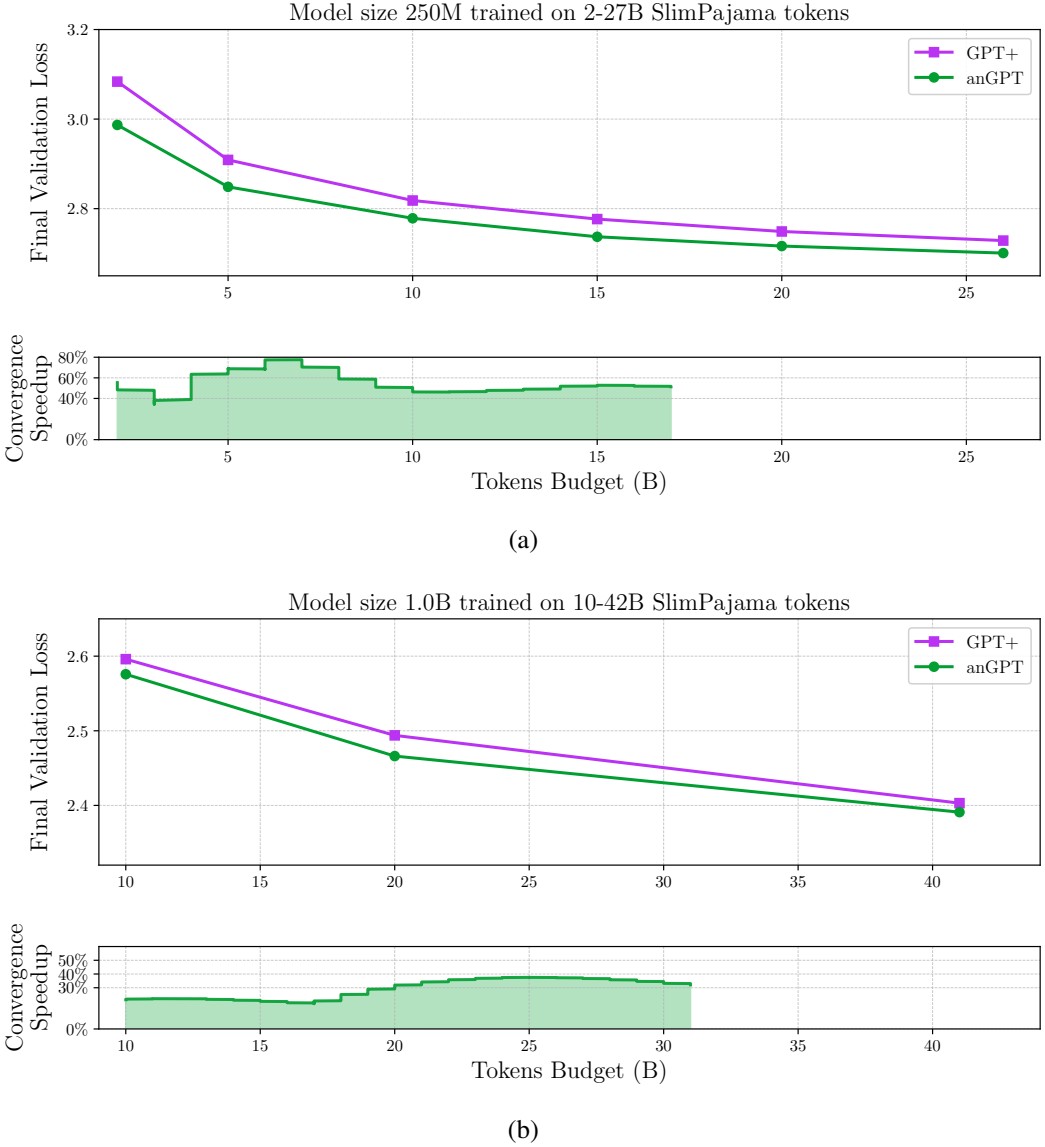

Figure I.1: Convergence comparison for (a) 250M and (b) 1B parameter models. Similar to the 0.5B results, anGPT achieves the same loss with significantly fewer iterations, demonstrating consistent 40% speedup across model scales.

## J  Derive scaling laws

For each combination of compute scale $C$ and architecture, we select a point with the minimal validation loss. We use the approach described in Hoffmann et al. [22]: we bin compute into 1500 FLOPs logarithmically spaced intervals and for each bin we obtain a point with the minimal loss. We obtain a mapping from each combination of parameters and number of tokens to the compute $C$. Following Hoffmann et al. [22], [36] we assume that the validation loss $\mathcal{L}(C)$ is proportional to $C^{-\alpha}$ ($\alpha > 0$) up to some positive constant $A_0$.

From the Figure 4 we see that GPT+ has a slightly higher coefficient $A_0$, which means it starts off with a higher loss than anGPT. Both GPT+ and anGPT have nearly identical estimated scaling law parameters, which implies that improvement of validation loss is **comparable** across these architectures.

We follow [22] and model compute optimal number of tokens $N_{opt}(C)$ and $D_{opt}(C)$ as power laws. From the set of point $\{N, D\}$ we select such $D$ and $N$ that correspond to minimal validation loss:

$$D_{opt}; N_{opt} := \arg\min \mathcal{L}(C)$$

To estimate 95% confidence intervals for model predictions, we propagate the uncertainty from the fitted parameters through the model. This involves computing the Jacobian matrix $J$ of the model output with respect to the parameters, evaluated at the extrapolated inputs. The variance of the predicted values is then approximated using the delta method as

$$\sigma^2 = J^\top \operatorname{Cov}(\hat{\theta}) \, J.$$

The resulting confidence intervals are given by

$$\hat{y} \pm t_{\alpha/2,\, n-p} \cdot \sigma,$$

where $t_{\alpha/2,\, n-p}$ is the critical value from the Student's $t$-distribution at a significance level of $\alpha = 0.05$. We fit power functions through the obtained points to estimate parameters for $D_{opt}(C)$ and $N_{opt}(C)$. The obtained power law fits are shown in Figure J.1, and the estimated parameters are in Table J.1.

| Model | $D_0$ | $\alpha_D$ | $N_0$ | $\alpha_N$ |
|-------|-------|-----------|-------|-----------|
| GPT+  | 13.408079 | 0.466271 | 0.336617 | 0.465600 |
| anGPT | 9.727337 | 0.473190 | 0.650506 | 0.451378 |

Table J.1: Estimated scaling law parameters for compute-optimal dataset size for $D_{\mathrm{opt}}(C)$ and compute-optimal model size $N_{\mathrm{opt}}(C)$. The table reports fitted values for both anGPT and GPT+. We observe that both models exhibit very similar exponents, which suggests **comparable trends in how optimal dataset and model size should be selected** under certain compute.

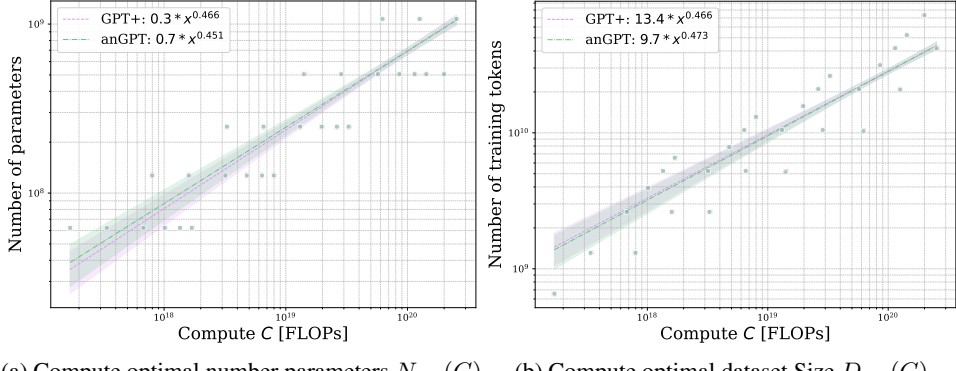

(a) Compute optimal number parameters $N_{opt}(C)$    (b) Compute optimal dataset Size $D_{opt}(C)$

Figure J.1: Scaling laws for compute-optimal model size and dataset size. a) shows the estimation for compute optimal number of parameters $N_{opt}(C)$ and b) shows compute-optimal dataset size $D_{opt}(C)$, both as function of compute $C$. We observe that both anGPT and GPT+ have similar scaling trends for compute-optimal allocations. Notably, both fits exhibit high uncertainty, which is indicated by the wide confidence intervals surrounding the curves.

# K    Downstream Task Evaluation

We evaluate our models on standard benchmarks to verify that pretraining improvements transfer to downstream tasks. Table K.1 shows results for 1B models across different training budgets.

Table K.1: Downstream evaluation on 1B models. Higher is better for all metrics except perplexity (PPL).

| Metric | Model | Training Budget | | |
| --- | --- | --- | --- | --- |
| | | 10B Tokens | 21B Tokens | 42B Tokens |
| PIQA (Acc) | GPT+ | 0.669 | 0.674 | 0.700 |
| | anGPT | **0.680** | **0.702** | **0.718** |
| ARC-Easy (Acc) | GPT+ | 0.508 | 0.532 | 0.561 |
| | anGPT | **0.539** | **0.561** | **0.596** |
| HellaSwag (Acc) | GPT+ | 0.338 | 0.363 | 0.400 |
| | anGPT | **0.354** | **0.386** | **0.413** |
| LAMBADA (PPL) | GPT+ | 36.915 | 24.932 | 16.726 |
| | anGPT | **28.780** | **17.577** | **14.679** |
| WikiText (PPL) | GPT+ | 22.648 | 21.661 | 17.621 |
| | anGPT | **20.518** | **17.734** | **16.175** |
| WinoGrande (Acc) | GPT+ | 0.522 | 0.536 | 0.554 |
| | anGPT | **0.530** | **0.559** | **0.563** |
| MMLU (Acc) | GPT+ | **0.232** | 0.239 | 0.243 |
| | anGPT | **0.232** | **0.241** | **0.256** |

anGPT consistently outperforms GPT+ across benchmarks, with particularly strong improvements in perplexity-based metrics (LAMBADA, WikiText) and reasoning tasks (ARC-Easy, HellaSwag). The performance gains correlate well with the validation loss improvements observed during pretraining.

# L  Ablation Studies

We conduct comprehensive ablation studies to investigate the sensitivity of anGPT to variations in normalization factors. Table L.1 shows the impact of different modifications to the normalization scheme on a 0.5B model trained on 10B tokens.

Table L.1: Ablation study on normalization factors. We report relative change to anGPT in parentheses (%).

| Configuration | Validation Loss | PPL |
|---|---|---|
| GPT+ | 2.677 (+1.46%) | 14.541 (+3.94%) |
| anGPT (baseline) | 2.638 | 13.990 |
| anGPT (const. norm factor ×0.5) | 2.647 (+0.34%) | 14.116 (+0.90%) |
| anGPT (const. norm factor ×2.0) | 2.642 (+0.12%) | 14.033 (+0.31%) |
| anGPT (all norm factors ×0.5) | 7.935 (+200%) | 2792.69 (+19k%) |
| anGPT (all norm factors ×2.0) | NaN | NaN |
| anGPT (no const. norm factors) | 2.667 (+1.09%) | 14.400 (+2.93%) |
| anGPT (no residual norm factor) | 2.719 (+3.04%) | 15.156 (+8.33%) |
| anGPT (no norm factors) | 2.718 (+3.02%) | 15.148 (+8.28%) |
| anGPT (no LERP) | 2.688 (+1.89%) | 14.700 (+5.08%) |
| anGPT (half token budget) | 2.755 (+4.42%) | 15.718 (+12.36%) |
| anGPT (double token budget) | 2.565 (-2.78%) | 13.000 (-7.07%) |

These results demonstrate that while the normalization factors are sensitive to large perturbations (50% or 200% scaling), the method remains robust to smaller estimation errors. The residual normalization factors are most critical, as their manipulation can cause training collapse. The 1.09% performance drop from removing constant normalization factors is significant when compared to the performance gap between anGPT and GPT+ (1.46%).

# M  Evolution of Learnable Parameters

We track the evolution of learnable interpolation parameters $\alpha_A$ (attention) and $\alpha_M$ (MLP) throughout training. Figure M.1 shows these parameters across different layers and model sizes. The increase in $\alpha$ values demonstrates that the model learns to increasingly incorporate new features from each block.

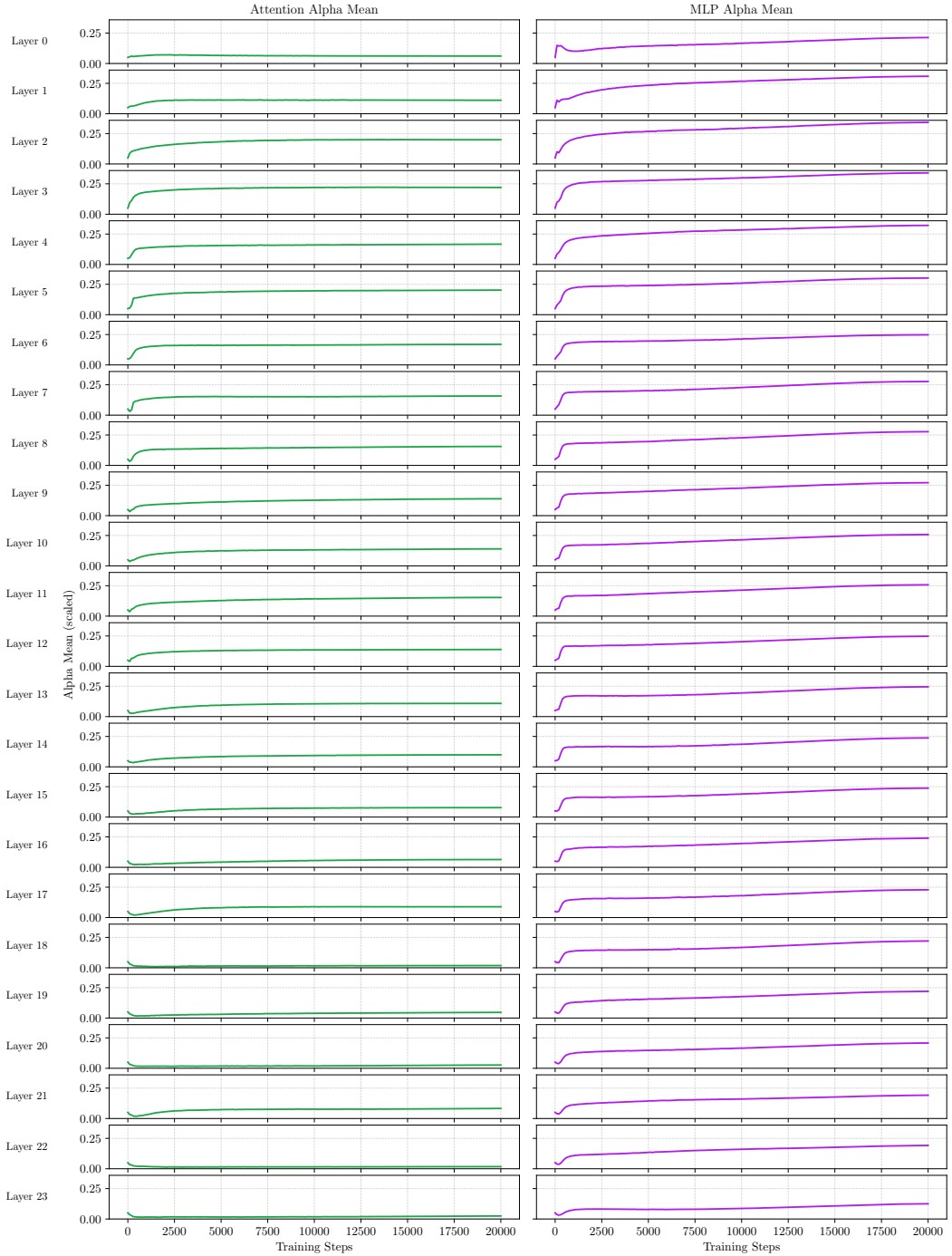

Figure M.1: Evolution of interpolation parameters during training of a 0.5B parameter model.

# N   Empirical Validation of Norm Concentration

To verify that approximate normalization maintains stable norms throughout the network, we measure the ratio of mean input norm to mean output norm for each block. Table N.1 shows these ratios at different layers.

Table N.1: Ratio of mean output norm to mean input norm for transformer blocks. Values close to 1.0 indicate stable norm propagation.

| Layer | anGPT | GPT+ |
|---|---|---|
| First attention | 1.07 | 0.01 |
| First MLP | 1.86 | 1.60 |
| Middle attention | 0.86 | 2.47 |
| Middle MLP | 0.97 | 2.60 |
| Last attention | 0.92 | 4.62 |
| Last MLP | 1.22 | 5.83 |

anGPT maintains near-unity ratios (0.86–1.86) while GPT+ exhibits significant norm growth in deeper layers (up to $5.83\times$), demonstrating the effectiveness of approximate normalization in stabilizing gradient flow. We measure the error introduced by using $\nu = 1/\sqrt{\mathbb{E}[\|x\|^2]}$ as an approximation:

Table N.2: Estimation error of normalization factors

| Factor | Relative Error |
|---|---|
| $\nu_{qkv}$ | 1.16% |
| $\nu_p$ | 0.07% |
| $\nu_{uz}$ | 0.02% |
| $\nu_d$ | 0.08% |
| $\nu_{acf}$ | 0.41% |
| $\nu(\alpha)$ | 0.00% |

All errors are below 2%, well within the robustness margins demonstrated in our sensitivity analysis.

# O   Robustness to Distribution Shift

We evaluate norm stability under out-of-distribution inputs by comparing normal text samples against pathological inputs (512 repetitions of the same token). This tests whether fixed normalization factors remain effective under extreme distribution shifts.

Table O.1: Mean residual norms under distribution shift (0.5B model)

| Model | Text Input | | Same Token | |
| --- | --- | --- | --- | --- |
| | Mean | Std | Mean | Std |
| GPT+ | 1831.42 | 353.93 | 545.68 | 78.80 |
| anGPT | 2.48 | 0.18 | 1.65 | 0.15 |

While both architectures exhibit norm changes under distribution shift, anGPT maintains better stability with only $1.5\times$ change compared to GPT+'s $3.4\times$ change. The moderate norms and absence of catastrophic failure indicate robustness to distribution shifts, supporting the generalizability of our approach beyond standard training distributions.

# P   Norms of the parameter groups in anGPT

Each line represents the average norm of the input direction, i.e., rows in case of $\boldsymbol{W}\boldsymbol{x}$.

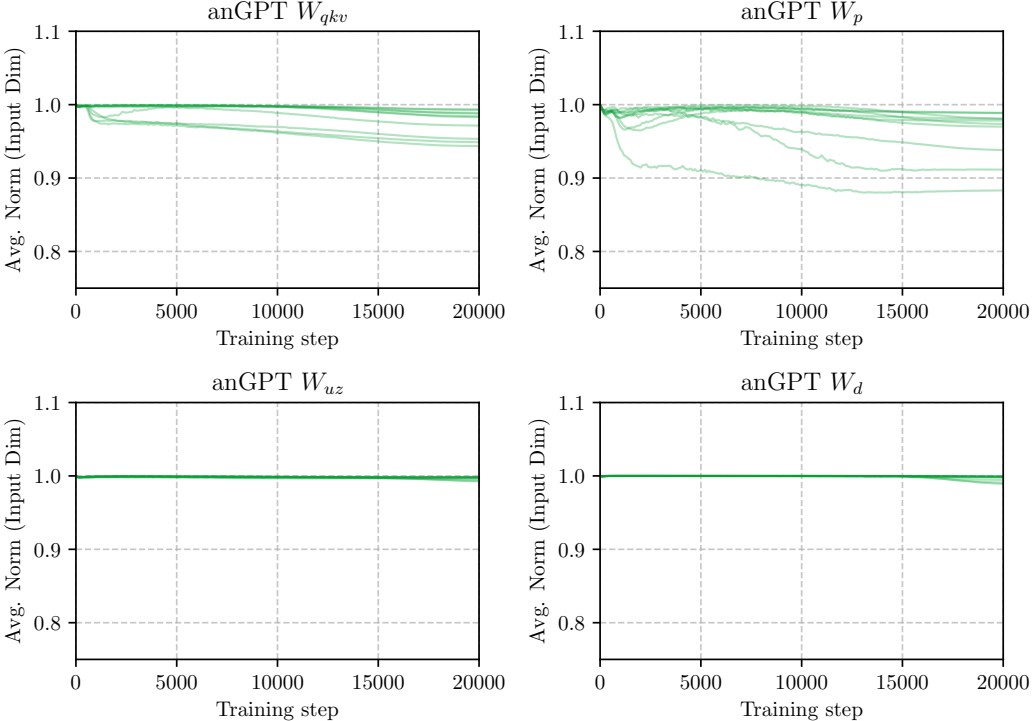

Figure P.1: The mean norm of dimension 1 (input dimension) of anGPT parameter groups during training. The traces correspond to individual layers. We see that despite the bound, the norm stays close to one.

