# OpenReview forum: "Learning in Compact Spaces with Approximately Normalized Transformer"
_NeurIPS.cc/2025/Conference — NeurIPS 2025 poster_

### Official Review · Reviewer_p2iW · 2025-06-28

**Clarity:** 3
**Significance:** 3
**Originality:** 3
**Rating:** 4
**Confidence:** 4

**Summary:**

Normalization is crucial for the training stability of deep neural networks. Previous work nGPT [1], extensively use normalization layers to update all latent representations on a hypersphere, which eliminates the need for pre-training techniques such as warmup schedules and weight decay. However, this extensive normalization introduces significant computational overhead. Based on concentration measure in high-dimensional spaces, this paper proposes a more efficient approach: rescale each feature dimension with precomputed normalization factors, thereby avoiding weight normalization, which makes the entire transformer model "approximately normalized".

[1] [nGPT: Normalized Transformer with Representation Learning on the Hypersphere](https://arxiv.org/abs/2410.01131)

**Questions:**

1. How do you ensure that core assumptions, such as the i.i.d. condition or features being uniformly distributed on the sphere, continue to hold after training?

2. Could you address the additional comments in the weaknesses section?

**Ethical Concerns:**

["NO or VERY MINOR ethics concerns only"]

**Final Justification:**

I would like to thank the authors for their detailed explanations. This work is valuable, and its main shortcomings pertain primarily to presentation, which can be easily addressed. I recommend that the authors present the derived normalizing factors in a dedicated table. Additionally, providing the exact effective learning rates explicitly in the appendix would enhance accessibility. Lastly, the paper would become even more convincing if framed as an strong empirical study motivated by loose theoretical insights rather than theory-driven with empirical support. While the theoretical rigor does not meet high standard, I agree that the proposed approach is practically useful.

**Limitations:**

Yes

**Paper Formatting Concerns:**

No concern regarding format.

**Quality:**

3

**Strengths And Weaknesses:**

**Strengths**

1. The motivation and core idea of this paper are clear: nGPT accelerates LLM pre-training and eliminates the need for hyperparameters related to learning rate schedules and weight decay. Given that feature vector norms concentrate in high-dimensional spaces, pre-estimating these norms allows normalization steps to be avoided.
2. The empirical study of the paper is comprehensive, featuring clear scaling trends and insightful ablation studies.

**Weaknesses**

1. The example regarding fast contraction in Section 3.1 is not clearly connected to the authors' arguments about global learning rate selection. It would be beneficial to either find a more directly relevant example or to clarify the existing connection.

2. The paper lacks some important details: (a) What are the exact procedures used to derive each of the normalization factors? (a) In Section 3.2, why $||h_{\leq L} || = \sum_{l=1}^L ||h_l || $? (c) How precisely are the "effective learning rates" applied? Although readers might guess, explicit clarification from the authors would significantly improve this paper.

3. While assumptions such as the i.i.d. condition clearly hold at initialization, the paper does not sufficiently explain why these assumptions remain valid throughout training.

---

> ### Author Rebuttal · Authors · 2025-07-30
>
> Dear Reviewer p2iW,
>
> Thank you for your review and feedback on our paper. We appreciate the positive assessment and the constructive feedback. We address our named weaknesses and answer your questions in the follow:
>
> > 1 The example regarding fast contraction in Section 3.1 is not clearly connected to the authors' arguments about global learning rate selection.
>
> For our example problem $\min_{x} \frac{1}{2} x^\top \Lambda x$ with $\Lambda = \text{diag}(\lambda_1, \dots, \lambda_d)$ and $\lambda_i > 0$, the contraction rate for the $i$th entry is given by $\rho_i(\alpha) = | 1 \alpha \lambda_i|$.
> We can make any specific entry $\lambda_i$ converge fast (in one step), if we set $\alpha = (\lambda_i)^{-1}$. However, other entries $\lambda_{j\neq i}$ may either converge slowly or diverge. The best (global) learning rate $\alpha^\star$ that achieves the fastest overall contraction while ensuring convergence for all entries is given by
> $\alpha^\star = \frac{2}{\lambda_{\max} + \lambda_{\min}}$, which achieves a contraction rate of
> $$\rho_{\max}(\alpha^\star)= \frac{\kappa - 1}{\kappa + 1} $$
> with $\kappa = \lambda_{\max} / \lambda_{\min}$. Hence, the speed of contraction is problem dependent and may be slow even with the knowledge of $\alpha^\star$ (best possible fixed learning rate).
> However, if we apply normalization before running gradient descent on this problem, we always have $\lambda_{\max} = \lambda_{\min}$ and thus fast convergence (since  $\kappa=1$, convergence in one step).
> Consequently, we always achieve the fastest possible convergence for $\alpha^\star$. Note that it does not matter how we found the optimal learning rate, *via manual tuning, grid search or sophisticated hyperparameter tuning*, but we argue that it is the most effective in a setting where $\lambda_{\max} = \lambda_{\min}$ (which normalization can help to achieve) as all variables may converge at similar rates.
> We will clarify this point in the paper.
>
> > 2a) What are the exact procedures used to derive each of the normalization factors?
>
> The procedure to derive the normalizing factors is decribed in Section 4.2: "Derivation of the normalizing factors". We use
>
> $\eta^{-1}_{g(x)} = \sqrt{\mathbb{E}[\Vert g(x) \Vert_2^2]}$
>
> for analytical computations in case of linear maps and the residual update. For the activation function, we use numerical computation using Monte Carlo estimation for the activation function to compute
>
> $\eta^{-1}_{acf} = \mathbb{E}[\Vert \text{acf}(x) \Vert_2]$.
>
> The respective normalizing factors are then given by $\nu = 1/\eta^{-1}$. We list the specific normalization factors we used below with model dimension $d$, head dimension $h$ ($64$ in our case) and feed-forward dimension $f$ (typically $4 \times d$) :
>
> $\nu_{qkv} = 1 / \sqrt{\frac{h}{d}} $ for $W_{qkv} \in \mathbb{R}^{h \times d} $
>
> $\nu_{p} = 1 $ for $ W_p \in \mathbb{R}^{d \times d} $
>
> $\nu_{uz} = 1 / \sqrt{\frac{f}{d}}$ for $ W_{uz} \in \mathbb{R}^{f \times d} $
>
> $\nu_{acf} = 1 / \text{Monte Carlo estimation}$
>
> $\nu_{d} = 1 / \sqrt{\frac{d}{f}}$ for $ W_{uz} \in \mathbb{R}^{f \times d} $
>
> We are sorry that this is not clear, we will clarify this in the paper accordingly.
>
> > 2b) In Section 3.2, why $\Vert h_{\le L} \Vert = \sum^{L}_{l=1} \Vert h_l \Vert $ ?
>
> We define $h_{\le L} := \sum_l^L h_l$, i.e., the residual state of the classic transformer after summing the contribution of $L$ blocks with, for simplicity, normalized outputs (post-norm). Estimating its norm should give us an idea about the growth of the hidden state in this case.
>
> Note that we analyze $ \mathbb{E}[\Vert h_{\le L} \Vert_2^2] $, not $\Vert h_{\le L} \Vert_2 = \sqrt{\Vert h_{\le L} \Vert_2}$ (or $\Vert h_{\le L} \Vert_2)$. For $\Vert h_{\le L} \Vert_2 $ (not squared) we cannot pull the expected value inside the $\sqrt{\cdot}$ operation. Consequently, $ \mathbb{E}[\Vert h_{\le L} \Vert_2] = \sum_{l=1}^L \mathbb{E}\Vert h_l \Vert $ would not be correct. However, we unfortunately only stated $\Vert h_l \Vert = 1 $ but forgot to add the assumption $\mathbb{E}[h_l] = 0$. Using these assumptions, the derivation of $\mathbb{E}[\Vert h_{\le L} \Vert_2^2]$ is as follows:
> \begin{align*}
> \Vert h_{\le L} \Vert_2^2 = \left( \sqrt{ h_{\le L}^\top h_{\le L} } \right)^2
> = h_{\le L}^\top h_{\le L}
> = \left(\sum_l h_l \right)^\top \left(\sum_l h_l \right) = \sum_l \sum_{l'} h_l^\top h_{l'}
> \end{align*}
> Now we analyze the expected value of this under the assumption that $h_l$ are independently distributed and $\mathbb{E}[h_l] = 0$. From the assumptions follows, that for $l\neq l'$, $ \mathbb{E} [h_l^\top h_{l'}] = 0$.
> Consequently,
> \begin{align*}
> \mathbb{E}[ \Vert h_{\le L} \Vert_2^2 ] &= \mathbb{E} \left[ \sum_l \sum_{l'} h_l^\top h_{l'} \right]
> &= \sum_l \sum_{l'} \mathbb{E} [ h_l^\top h_{l'} ] = \sum_l \mathbb{E} [ h_l^\top h_{l} ]
> &= \sum_l \mathbb{E}[ \Vert h_l \Vert_2^2] = L
> && \text{since for $l \neq l'$, $ \mathbb{E} [h_l^\top h_{l'} ] = 0$ }  \text{and $\Vert h_l \Vert = 1$ by assumption}
> \end{align*}
>
> > 2c) How precisely are the "effective learning rates" applied? Although readers might guess, explicit clarification from the authors would significantly improve this paper.
>
> We adopted the effective learning rates from the nGPT [1] paper but in our work crucial implementation detail deserves explicit clarification.
> Following nGPT's approach, for any trainable scaling parameter $ s_a$ (such as $\alpha_A$, $ \alpha_M$, $s_z$, etc.), we employ a reparameterization scheme using two scalars: $ s_{a,\text{init}}$ and $s_{a,\text{scale}}$. Instead of directly optimizing $s_a$, we optimize a surrogate parameter $ \hat{s}_a$ and compute:
>
> $$ s_a = \frac{s_{a,\text{init}}}{s_{a,\text{scale}}} \cdot \hat{s}_a $$
>
> The surrogate parameter is initialized as:
>
> $$ \hat{s_a}^{(0)} = s_{a,\text{scale}}$$
>
> which ensures the initial value is:
>
> $$ s_a^{(0)} = \frac{s_{a,\text{init}}}{s_{a,\text{scale}}} \cdot s_{a,\text{scale}} = s_{a,\text{init}}$$
>
> Following the standard Adam update rule:
>
> $$ \hat{s}_a^{(t+1)} = \hat{s}_a^{(t)} - \alpha \cdot \frac{m^{(t)}}{\sqrt{v^{(t)}} + \epsilon}  $$
>
> where $ m^{(t)} $ and $ v^{(t)} $ are the momentum terms computed using the scaled gradient. The effective learning rate for the actual parameter $s_a$ becomes:
>
> $$ \alpha_{\text{eff}} = \alpha \cdot \frac{s_{a,\text{init}}}{s_{a,\text{scale}}}$$
>
> This allows us to control the learning dynamics of different parameter types while maintaining a single global learning rate $\alpha$. For the interpolation parameters $\alpha_A$ and $\alpha_M$:
>
> $ s_{\alpha,\text{init}} = 0.01, \quad s_{\alpha,\text{scale}} = \frac{1}{\sqrt{d_{\text{model}}}}$
>
> This yields an effective learning rate:
>
> $ \alpha_{\text{eff}}(\alpha_A, \alpha_M) = \alpha \cdot 0.01 \cdot \sqrt{d_{\text{model}}}$
>
> For scaling parameters like $s_z$:
>
> $ s_{z,\text{init}} = 1, \quad s_{z,\text{scale}} = \frac{1}{\sqrt{d_{\text{model}}}}$
>
> yielding: $ \alpha_{\text{eff}}(s_z) = \alpha \cdot \sqrt{d_{\text{model}}}$
>
> This mechanism addresses the fundamental challenge that different parameter types require different update magnitudes. Our reparameterization ensures that parameters with magnitude $ \mathcal{O}(1/\sqrt{d_{\text{model}}})$ receive appropriate learning rates, all parameters can be optimized with a single global learning rate, and training remains stable by avoiding the need for manual learning rate scheduling across parameter types. This approach ensures that scaling parameters like $s_z$ are updated with effective learning rates comparable to other normalized parameters in the network, eliminating the need for separate learning rate schedules. We will add more details to Section 4.3.
>
> > 3 While assumptions such as the i.i.d. condition clearly hold at initialization, the paper does not sufficiently explain why these assumptions remain valid throughout training. / How do you ensure that core assumptions, …, continue to hold after training?
>
> The theory only serves as motivation to investigate the application of the norm concentration phenomenon empirically. The assumptions do not hold in training, partially by design. As stated in Section 4.1 line 140: “While these assumptions may not hold during training, we may still observe concentration empirically, particularly due to the high dimensionality of representations”. Also, it does not hold since the weights in our case are not normalized but only bounded, see Sec. 4.2  Constraining Parameters. Bounding the norm of the parameters instead of normalizing led to better results empirically, see Fig. 5. Additionally, one can see from Figure 1, that the intermediate representations are not truly normalized (but only approximately) reaching norms up to ~2.9.
>
> Due to the post-norm operation and our approx. normalization factors, we observe many of the beneficial properties associated with normalization such as a very moderate growth of the norm on the residual connection (Fig. 1).
>
> Experimentally, we can see that the blocks in anGPT keep the normalization concentration of norm due to the norm factors. The ratio of the mean input norm to the mean output norm of a block (for anGPT before the post norm and for GPT+ after the RMS Norm) on different layer (first, middle and last) does deviate less in anGPT than in GPT+ Please find the results of this experiment in the Table in Section 1.2 in the response to Reviewers skhp.
>
> Thank you again for the thorough and constructive review. Your feedback will help us improve our work. Please mind, we also performed downstream evaluations and additional ablation experiments, see response to Reviewer SY5g. Please do not hesitate to ask further questions. In case we addressed your concerns, we would appreciate a reconsideration of your score.
>
>
> Best regard, the authors.
>
> ___
> [1] Loshchilov, Ilya, et al. "ngpt: Normalized transformer with representation learning on the hypersphere." 13th International Conference on Learning Representations (ICLR 2025) .

---

### Official Review · Reviewer_skhp · 2025-06-30

**Clarity:** 3
**Significance:** 2
**Originality:** 3
**Rating:** 4
**Confidence:** 3

**Summary:**

The paper introduces the approximately normalized Transformer (anGPT), a variant of the Transformer architecture that replaces standard normalization layers (e.g., LayerNorm) with fixed, pre-computed scaling factors. The central premise is that in high-dimensional spaces, the norms of random vectors tend to concentrate tightly around their expected values, a phenomenon known as concentration of measure. Leveraging this property, anGPT applies fixed scalar normalization factors throughout the network to approximate norm preservation, while also constraining the weight matrices to lie within a compact (bounded-norm) parameter space. This design eliminates the need for learnable normalization parameters, weight decay, and learning rate warm-up, resulting in faster convergence with minimal computational overhead.

**Questions:**

I appreciate the effort to simplify normalization in large Transformer models through a theoretically motivated framework. However, several mathematical and empirical claims in the paper warrant deeper scrutiny. Below are the specific questions intended to clarify the assumptions and implications of the proposed method.

---
1) The core approximation in anGPT assumes that for high-dimensional inputs \( x \), the output norm of a function \( g(x) \) concentrates:
\[
\|g(x)\|_2 \approx \mathbb{E}[\|g(x)\|_2]
\]
This is motivated by classical concentration results (e.g., Vershynin's theorem for \( x \sim \text{Unif}(\mathbb{S}^{d-1}) \)).

- How valid is this assumption during training, where the input distribution evolves due to gradient updates and may be far from uniform or isotropic?
- Can you provide empirical evidence of concentration in intermediate activations (e.g., histograms of \( \|g(x)\|_2 \))?
---
2) anGPT replaces exact normalization with a scalar factor \( \nu = 1 / \sqrt{\mathbb{E}[\|x\|^2]} \), which is a biased approximation of \( \mathbb{E}[1/\|x\|] \).

- What is the theoretical or empirical impact of this bias on gradient magnitudes or effective learning rates?
- Does this approximation systematically distort the optimization trajectory or loss surface?
- Can you quantify the tradeoff between computational efficiency and scaling accuracy introduced by this simplification?

---
3) anGPT modifies the residual connection using a learnable interpolation:
\[
h \leftarrow (1 - \alpha) h + \alpha x
\]
with an additional normalization factor based on \( \nu(\alpha) \).

- This changes the gradient propagation structure. Have you analyzed the Jacobians \( \partial h / \partial x \) and \( \partial h / \partial h_{\text{prev}} \) to assess gradient flow?
- For small \( \alpha \), does the network risk under-utilizing new features, leading to a form of representational inertia?
---

4) The effectiveness of fixed normalization factors implicitly assumes consistent norm statistics across training and test distributions.

- How does anGPT perform under domain shift, few-shot adaptation, or continual learning settings where norm distributions may vary?
- Have you tested whether norm concentration persists in out-of-distribution scenarios?
- Could fixed normalization factors cause feature collapse or instability in low-data or transfer regimes?

**Ethical Concerns:**

["NO or VERY MINOR ethics concerns only"]

**Final Justification:**

Initially, I was concerned about the concentration of measure phenomena during training and the method’s robustness to distribution shifts. However, the authors have addressed my questions thoroughly. I now believe this is a valuable piece of research that contributes to accelerating the training and convergence of large language models.

**Limitations:**

Yes

**Quality:**

3

**Strengths And Weaknesses:**

# Strengths

1) By replacing LayerNorm with fixed scalar normalization factors, anGPT reduces the computational and memory overhead typically associated with normalization, especially during inference, without sacrificing stability.

2) anGPT achieves up to 40% faster convergence compared to GPT models with QK normalization, while removing the need for learning rate warm-up and weight decay, thus simplifying hyperparameter tuning.
# Weaknesses

Please refer to the detailed questions I provided below, which highlight potential concerns around:

* The validity of norm concentration under non-uniform, evolving training distributions.
* The bias introduced by using fixed normalization factors.
* The effect of LERP residuals on gradient flow.
* Unclear robustness to domain shifts and out-of-distribution data.

---

> ### Author Rebuttal · Authors · 2025-07-30
>
> Dear Reviewer skhp,
>
> Thank you for your extensive review of our paper. We appreciate the good rating for quality, clarity, and originality. In the following we will respond to your detailed questions.
>
> But first of all, we would like to comment on your summary by pointing out that the theory only serves to motivate the empirical investigation of the phenomenon, and, if it can help to gain some of the benefits that have been empirically observed when using normalizations, see nGPT[1]. Our goal is ultimately to improve transformers for language modelling similar to nGPT but using cheap computations. For this, we show extensive pretraining results and derive scaling rules for our proposed architecture.
> Regarding your named strength,
>
> > …anGPT reduces the computational and memory overhead typically associated with normalization…
>
> we would like to point out that our method does not reduce memory overhead nor computational overhead in contrast to standard GPT architecture, but we do reduce overhead in contrast to nGPT [1], see Table 2.
>
> > 1.1 How valid is this assumption during training, where the input distribution evolves due to gradient updates and may be far from uniform or isotropic?
>
> As stated in Section 4.1 line 140: “While these assumptions may not hold during training, we may still observe concentration empirically, particularly due to the high dimensionality of representations”. This can also be seen from Figure 1, where the norms of the residual state vectors do not display a norm of one.
>  We even violate the assumptions directly by using bounded instead of normalized weights since we observed it to produce better results empirically (see Ablation study in Fig. 5 “nGPT with param bounding”).
>
> > 1.2 Can you provide empirical evidence of concentration in intermediate activations (e.g., histograms of ( |g(x)|_2 ))?
>
> Thanks for this question, we missed to show this in our submission. Figure 1 shows only the residual norm including the post normalization. In addition, we can measure the norm change of a block (g(x) = MLP() or Attention() before the post norm). Since we can’t add diagrams to the rebuttal we report the ratio of the mean input norm to a block to the mean output norm of a block (for anGPT before the post norm and for GPT+ after the RMS Norm) on different layer (first, middle and last).
>
> | | anGPT | GPT+ |
> |:---|---:|---:|
> | first attn | 1.07 | 0.01 |
> | first mlp | 1.86 | 1.60 |
> | mid attn | 0.86 | 2.47 |
> | mid mlp | 0.97 | 2.60 |
> | last attn | 0.92 | 4.62 |
> | last mlp | 1.22 | 5.83 |
>
> We see that the approximate normalization factors stabilize the norm of a block without explicit normalization by keeping the ratio close to 1. We will add this analysis including a histograms plot to the paper.
>
> > 2.1 What is the theoretical or empirical impact of this bias on gradient magnitudes or effective learning rates?
>
>
> Since we use the square root of estimates for the squared norm as scaling factors (for the linear maps and the interpolation on the residual connection) we can expect an overestimation of the norm due to Jensen's inequality (as stated in Sec. 4.2), as
> $\sqrt{\mathbb{E}[ \Vert x \Vert^2]} \ge \mathbb{E}[ \Vert x \Vert]$
> due to the concavity of the $\sqrt{\cdot}$. Consequently, we expect and underestimation of the inverse of $ 1/ \mathbb{E}[ \Vert x \Vert ] $, which is again an underestimation of $\mathbb{E}[1 / \Vert x \Vert]$ due the the convexity of $1/x$.
> Overall, we may expect an underestimation of the inverse norm which should result in norms larger than 1. We also added this to Section 4.2 for the derivation of the scaling rules.
> Since these scaling factors scale the weight matrix/activations, they also scale gradients in the same way. Hence, gradients may be scaled up based on the degree of underestimation (due to Jensen's inequality).
>
> We also measured the empirical impact by performing additional ablation experiments, please find all results in Table 1 in the response to Reviewers SY5g.
> When we remove the constant normalization factors, we face a validation loss increase of 1.09%, if we remove all normalization factors (including on the residual) we see an increase of 3%, similar to removing only the normalization factor on the residual connection. If we replace all normalization factors by exact normalization ($1/||x||_2$) we get a slightly worse validation loss of +0.4% and an increased runtime of 30%.
>
> We hypothesize that this slight loss increase is caused by having normalized, instead of bounded representations, in line to the observation in Fig. 5, where using bounded weights
> instead of normalized weights improve performance. We add this analysis to the paper.
>
>
>
>
> > 2.2 Does this approximation systematically distort the optimization trajectory or loss surface?
>
> As outlined in the quadratic toy example, normalization may distort the loss landscape for problems with axis-aligned distortions, in other words, if some parameters are more sensitive than others.
>
> > 2.3 Can you quantify the tradeoff between computational efficiency and scaling accuracy introduced by this simplification?
>
> We calculated the error of computing $\nu = 1 / \sqrt{\mathbb{E}[\Vert x \Vert^2]}$ (using MC estimation with 10^5 samples), which is a biased approximation of ($ \mathbb{E}[1/\Vert x\Vert ]$ ):
> | | $abs(\nu - \nu_{MC})$ |
> |:--|--:|
> | $\Delta \nu_{qkv}$  | 0.047 (1.16%) |
> | $\Delta \nu_{p}$  | 0.001 (0.07%) |
> | $\Delta \nu_{u}$  | 0.000 (0.02%) |
> | $\Delta \nu_{d}$  | 0.002 (0.08%) |
> | $\Delta \nu_{acf}$  | 0.015 (0.41%) |
> | $\Delta \nu_{alpha}$  | 0.000 (0.00%) |
>
> Based on our sensitivity analysis in response to Reviewers SY5g, we do not expect any negative impact caused by these low error values.
>
> > 3.1 This changes the gradient propagation structure. Have you analyzed the Jacobians to assess gradient flow?
>
> We did not perform this analysis but adapted linear interpolation on the residual from nGPT [1]. While we think the idea is interesting, we are not exactly sure what we could learn from analyzing the Jacobians (assessing better from worse).
>
> > 3.2 For small ($\alpha$ ), does the network risk under-utilizing new features, leading to a form of representational inertia?
>
> The factors $\alpha$ are learnable and were adopted from nGPT [1]. In general, our architecture can also be adapted to using standard residuals updates (See Section 4.2 for the respective scaling factors), but using LEARP performed better in our experiments.
> We performed an ablation where we removed the LERP residual connection and find a validation loss increase of 1.89% (for reference, using GPT+ instead of anGPT increase validation loss of 1.46%). Please find the numbers in Table 1 in Reviewers SY5g rebuttal.
>
> Additionally, we logged the $\alpha$ during training and see an increase from an initial value of 0.05 to 0.12 -0.25 till the end of the training across all layers and multiple model sizes. This could indicate that the model learns to increase $\alpha$ since the block output is beneficial for the model performance. We argue that we do not face an under-utilizing or representational inertia due to the increased performance in contrast to GPT+.
> We will add this experiment and analysis to the paper.
>
> > 4.1 How does anGPT perform under domain shift, few-shot adaptation, or continual learning settings where norm distributions may vary?
>
> This work focuses on pretraining and the introduction of architecture and deriving scaling laws. The empirical investigation of cases like few-shot adaptation or continual learning is out of the scope of this work.
> Generally, it is important to note that the distribution assumptions, such as uniform on a sphere, are required for the distribution of the *entries in the representation vector or weight matrixes, not between tokens*. The distributions of the tokens do not directly influence the normalization factors. We thus do not expect any out of the ordinary problems when faced with domain shift or continual learning.
>
> > 4.2 Have you tested whether norm concentration persists in out-of-distribution scenarios?
>
> Since the normalization factors do not use the data distribution in their derivation and are also not adapted throughout training, one may assume that there is no out of the ordinary impact when faced with out-of-distribution scenarios.
> Nevertheless, we performed an experiment where we forward passed with a 0.5B model 1000 text samples(wikitext-2) in contrast to 1000 samples containing 512 times the same token and analyzed the mean norm of the residual connection.
>
> | | mean | std |
> |:---|---:|---:|
> | text input GPT+  | 1831.42 | 353.93 |
> | same token GPT+  | 545.68 | 78.80 |
> | text input anGPT | 2.48 | 0.18 |
> | same token anGPT | 1.65 | 0.15 |
>
> We observe a clear change in the norm using an out-of-distribution input, however the change is for anGPT ($\times 1.5$) smaller as for GPT+ ($\times 3.4$). Notably, the residuals norms still stay moderate and there is no indication of a severe breakdown.
>
> > 4.3 Could fixed normalization factors cause feature collapse or instability in low-data or transfer regimes?
>
> Since the derivation of the normalization factors did not consider the data distribution, we do not expect any out of the ordinary impact of data-dependent artifacts coming from low-data distributions or transfer regimes.
>
> Thank you again for the thorough and constructive review. Your questions and  feedback helped us improve our work. Also, we added more ablation experiments, an analysis to the expected error of the normalization factor and down stream evaluations (see response to Reviewer SY5g). Please do not hesitate to ask further questions. In case we addressed your concerns, we would appreciate a reconsideration of your score.
>
> Best regard, the authors.
>
> ___
>
>
> [1] Loshchilov, Ilya, et al. "ngpt: Normalized transformer with representation learning on the hypersphere." 13th International Conference on Learning Representations (ICLR 2025) .

---

> ### Comment · Reviewer_skhp · 2025-08-06
>
> I thank the authors for their comprehensive rebuttal. I initially had major concerns regarding the concentration of measure phenomena during training and potential distribution shifts. However, the authors addressed these issues thoroughly, and I am satisfied with their responses. Therefore, I am increasing my score to 4.

---

### Official Review · Reviewer_7NrH · 2025-07-03

**Clarity:** 4
**Significance:** 3
**Originality:** 2
**Rating:** 5
**Confidence:** 1

**Summary:**

The paper introduces the Approximately Normalized Transformer (anGPT), a modification of the GPT architecture designed to stabilize training and accelerate convergence without relying on traditional normalization layers or extensive hyperparameter tuning. By using approximate normalization through input-independent scaling factors, anGPT mitigates the "Curse of Depth," where deeper layers in vanilla GPT models experience growing input norms, leading to training instability. The approach eliminates the need for weight decay and learning rate warm-up, reducing hyperparameter complexity. Experimental results demonstrate that anGPT achieves up to a 40% convergence speedup compared to the vanilla GPT, with only a minimal (~3%) increase in training step runtime and potentially reduced inference times. The authors validate their approach on model sizes from 32M to 1B parameters and provide detailed hyperparameter scaling trends and compute-dependent scaling laws.

Contributions:

1- Proposes the anGPT architecture, which uses approximate normalization to stabilize training and improve convergence speed.

2- Eliminates the need for weight decay and learning rate warm-up, simplifying hyperparameter tuning.

3- Demonstrates up to 40% faster convergence compared to vanilla GPT, with minimal runtime overhead.

4- Provides comprehensive empirical analysis, including hyperparameter scaling trends and compute-optimal scaling laws.

**Questions:**

see Strengths And Weaknesses

**Ethical Concerns:**

["NO or VERY MINOR ethics concerns only"]

**Final Justification:**

Authors addressed my comments, I am keeping my score at 5

**Limitations:**

Yes

**Quality:**

3

**Strengths And Weaknesses:**

Strengths:

- Novel Approach to Normalization: The paper introduces a principled and lightweight alternative to full normalization layers using approximate scalar normalization factors derived from concentration of measure, which is both theoretically motivated and practically effective.

- Improved Efficiency: anGPT achieves significant convergence speedups (up to 40%) with only ~3% added training cost and no increase in inference time, demonstrating clear practical benefits.

- Simplified Training Pipeline: By eliminating the need for weight decay and learning rate warm-up, the method reduces hyperparameter tuning complexity, which is valuable in large-scale pretraining.

- Strong Empirical Validation: The authors conduct thorough experiments across multiple model sizes and data scales, including compute-optimal scaling laws, showing consistent performance improvements.

- Insightful Analysis: The work provides a clear theoretical explanation of the issues with norm growth in deep transformers (“Curse of Depth”) and demonstrates how approximate normalization addresses them.

Weaknesses:

- Limited Scope of Evaluation: The experiments are confined to GPT-style language models and two datasets. It remains unclear how the approach generalizes to other architectures (e.g., encoders) or modalities (e.g., vision, audio).

- Approximation Validity Not Fully Explored: While the use of norm concentration is well-motivated, the paper does not deeply investigate cases where this assumption may break down (e.g., during early training or low-dimensional representations).

- Complexity of Implementation: Despite simplifying training, the architectural modifications (LERP, ν-scaling factors) and need to derive dimension-specific constants may increase implementation burden for practitioners unfamiliar with the theory.

---

> ### Author Rebuttal · Authors · 2025-07-30
>
> Dear eviewer 7NrH,
>
> Thank you for your review and feedback on our paper. We deeply appreciate the positive assessment and address our named weaknesses in the follow:
>
> > Limited Scope of Evaluation: The experiments are confined to GPT-style language models and two datasets. It remains unclear how the approach generalizes to other architectures (e.g., encoders) or modalities (e.g., vision, audio).
>
> We are aware of this limitation (see Section 6) and see this as part of future work. In this work we focused on proper evaluation for LLM pretraining and on deriving scaling laws to provide empirical evidence for the benefit of approximate normalization.
>
> > Approximation Validity Not Fully Explored: While the use of norm concentration is well-motivated, the paper does not deeply investigate cases where this assumption may break down (e.g., during early training or low-dimensional representations).
>
> The assumptions break already in the early training, see Figure 1 in the middle, and are even violated by design since our weights are only bounded instead of normalized.
> We state this in Section 4.1 line 140: “While these assumptions may not hold during training, we may still observe concentration empirically, particularly due to the high dimensionality of representations”.
>
> But due to the post-norm operation and our approx. normalization factors, we observe many of the beneficial properties associated with normalization such as a very moderate growth of the norm on the residual connection (Fig. 1). We found an increased convergence speed of ~40% across all model dimensions from 256 to 1280. However we only plotted it for 0.5B since we trained this model size for longer. We will clarify that and add more plots like Figure 3 on different model scales to the appendix.
>
>
> > Complexity of Implementation: Despite simplifying training, the architectural modifications (LERP, ν-scaling factors) and need to derive dimension-specific constants may increase implementation burden for practitioners unfamiliar with the theory.
>
> We acknowledge this tension between architectural modifications and ease of use. Our work focuses on large scale training where a training run could cost thousands of dollars. The anGPT approach could save significant cost and energy which may justify the additional implementation overhead.
>
> Regarding your confidence rating of 1 ("educated guess"), we would like to clarify any technical details that might help increase confidence in the assessment. The core mathematical foundation (concentration of measure) is well-established, and our empirical validation follows rigorous scaling law methodologies from recent literature.
>
> Thank you again for the thorough and constructive review. Please mind, we also performed downstream evaluations and additional ablation experiments, see our response to Reviewer SY5g. Your feedback will help us improve both the current work and guide future research directions.
>
> Best regard, the authors.

---

### Official Review · Reviewer_SY5g · 2025-07-03

**Clarity:** 3
**Significance:** 3
**Originality:** 3
**Rating:** 4
**Confidence:** 4

**Summary:**

This paper proposes the approximately normalized Transformer (anTransformer), a novel architecture that introduces approximate normalization through fixed scalar multipliers, motivated by the concentration of measure in high dimensions. Instead of exact normalization, it constrains the norm of parameters and representations to reduce the need for components like weight decay and learning rate warm-up. Applied to a GPT-style model (anGPT), the proposed method achieves up to 40% faster convergence than GPT+ (with QK norm) and matches or outperforms nGPT, while incurring only 3% extra training cost and negligible inference overhead.

**Questions:**

- How sensitive is the method to the choice of fixed normalization factors? Would small misestimations hurt performance significantly?
- Did the authors evaluate the method on any downstream tasks or instruction tuning settings?
- Could the proposed approximate normalization be combined with other efficiency improvements, such as low-rank or quantized attention layers?

**Ethical Concerns:**

["NO or VERY MINOR ethics concerns only"]

**Final Justification:**

- Authors provided new downstream evaluations, which show consistent improvements over baselines.
- Ablations demonstrate moderate robustness to normalization factor variation.
- Theoretical motivations remain partially validated, and failure mode analysis is limited.
- Evaluations focus on a single model size and selected benchmarks.
- I maintain my borderline accept score, recognizing the practical promise while noting open questions around generalization and theoretical grounding.

**Limitations:**

yes

**Quality:**

3

**Strengths And Weaknesses:**

The paper is well-written and presents a compelling motivation for approximate normalization based on theoretical insights and empirical observations. It introduces a clean and practical design that removes the need for additional normalization layers while retaining stability and training efficiency. Experiments are thorough, covering scaling laws, ablations, and comparisons across multiple model sizes and compute budgets. The results show consistent convergence gains and demonstrate that anGPT can match or outperform both GPT+ and nGPT.

However, some concerns remain. First, the assumptions behind norm concentration and fixed normalization factors are only partially validated in practice; more ablations or analysis of failure modes would strengthen the argument. Second, while convergence is improved, the validation loss differences in the scaling law experiments are relatively modest. Lastly, downstream task evaluations are missing, which makes it unclear whether the improvements transfer beyond pretraining loss.

---

> ### Author Rebuttal · Authors · 2025-07-30
>
> Dear Reviewer SY5g,
>
> Thank you for your review and feedback on our paper. We appreciate the positive assessment and the constructive feedback. We address our named weaknesses and answer your questions in the follow:
>
> > First, the assumptions behind norm concentration and fixed normalization factors are only partially validated in practice
>
> The theory only serves as motivation to investigate the application of the norm concentration phenomenon empirically.
> As we state this in Section 4.1 line 140: “While these assumptions may not hold during training, we may still observe concentration empirically, particularly due to the high dimensionality of representations”. This violation is partially by design,  as the weights in our case are not normalized but only bounded, see Sec. 4.2  Constraining Parameters. Bounding the norm of the parameters instead of normalizing led to better results empirically, see Fig. 5.
> Additionally, one can see from Figure 1, that the intermediate representations are not truly normalized (but only approximately) reaching norms up to ~2.9. Nevertheless, we observe many of the beneficial properties associated with normalization such as a very moderate growth of the norm on the residual connection.
>
> > more ablations or analysis of failure modes would strengthen the argument. / How sensitive is the method to the choice of fixed normalization factors? Would small misestimations hurt performance significantly?
>
> We agree with the reviewer that additional ablations and analysis would be beneficial. Therefore, we run multiple additional experiments, please find all results in Table 1 below.
>
> When multiplying the const. norm factors with 0.5x and 2x we find a validation loss increase of 0.34% and 0.12% respectively. If we apply the same multiplier to the residual normalization factor ($\nu(\alpha)$ the training collapses. This is caused by exploding or vanishing latent representation on the residual $h$. However, when we remove the constant normalization factors, we face a validation loss increase of 1.09%, if we remove all normalization factors (including on the residual) we see an increase of 3%, similar to removing only the normalization factor on the residual connection.
>
> To put the validation loss improvement of 1.09% into perspective, by using GPT+ instead of anGPT the validation loss increases by 1.46%. When using a half / double token budget training anGPT it leads to an 4.42% increase / 2.78% decrease of the validation loss. So 1.09% display already a significant performance drop.
>
> So the normalization factors are sensitive but even with a 0.5x or 2x it is better than a GPT+ model.
> We expect small errors in the estimation of the normalization factors Jensen’s inequality (see response to Reviewer skhp to question 2.3) below 2% which is way below the tested 50% or 200 % and therefore we do not expect a significant negative impact of this error.
>
>
> If we manipulate the residual, we collapse the training and the normalization factors on the residual are most important. Using our calculated normalization factors on all proposed positions leads to the best performance.
>
> Regarding the failure modes, we would like to note that without any assumptions, anGPT would simply be a transformer with post norm, rescaled activations, and a missing normalization before the head layer that is trained with bounded weights. The missing normalization layer before the head could be added as a failsafe at minimal cost. On the other hand, bounding the weights can be seen as an alternative to weight decay, see [2].
> Consequently, none of this would lead us to believe that catastrophic failures may occur.
>
> We will add this clarification, the additional ablation experiments and the error analysis into the paper.
>
>
> > Second, while convergence is improved, the validation loss differences in the scaling law experiments are relatively modest.
>
> While in the submitted paper we only plot the convergence speedup for the 0.5B model we do have a similar convergence speedup (~40%) for the other model size including our largest trained model,with 1B parameters. We will add plots in the same style as Figure 3 for 1B and 247M into the appendix in the CRC. This consistent benefit is also reflected in the derived scaling laws of GPT+ and anGPT with a significantly smaller scale for anGPT, see Figure 4.
>
> In addition to the convergence speedup, we also eliminate the need for weight decay and learning rate warmup, and consequently the associated hyperparameter tuning effort.
>
> > Downstream task evaluations are missing, which makes it unclear whether the improvements transfer beyond pretraining loss. / Did the authors evaluate the method on any downstream tasks or instruction tuning settings?
>
> Upon your request, we performed downstream evaluations on all our largest trained models (1B) and find a performance improvement similar to the reduced validation loss (see Table 2 below). We outperform on six benchmarks and on all token budgets GPT+. Regarding the selection of the benchmarks, for some other benchmarks the models are around random guess or only slightly above (see MMLU), so we picked downstream tasks which give a signal at all. The weakness of the model is also the reason why we did not perform instruction tuning experiments.
>
> >Could the proposed approximate normalization be combined with other efficiency improvements, such as low-rank or quantized attention layers?
>
> Yes. The approximate normalization approach is orthogonal to low-rank or quantization attention. It could also be combined with other modifications such as linear attention or in hybrid architectures. We see the benefit of anGPT is the reduced computational overhead, which matters not only during training but also at inference of foundation models.
>
>
> Thank you again for the thorough and constructive review. The feedback will help us improve both the current work and guide future research directions. We added more ablation studies, an analysis to the expected error of the normalization factor and down stream evaluations.
> Please do not hesitate to ask further questions. In case we addressed your concerns, we would appreciate a reconsideration of your score.
>
> Best regards, the authors.
>
> ____
> **Table 1 Additional Ablation**
>
> This table contains additional ablation experiments on the normalization factor using a 0.5B model on 10B tokens. * We report the relative change to anGPT in (*%).
>
> |  | validation loss | PPL |
> | :---- |-----: | ----: |
> | GPT+       |   2.677 (1.46%)   | 14.541 (3.94%) |
> | anGPT*           |   2.638    | 13.990  |
> | anGPT (const. norm factor x0.5)   |  2.647 (0.34%)   | 14.116 (0.90%) |
> | anGPT (const. norm factor x2.0) |  2.642 (0.12%)   | 14.033 (0.31%) |
> | anGPT (all norm factor x0.5)   |  7.935 (200%)   | 2792.69 (19k %) |
> | anGPT (all norm factor x2.0) |  NaN   | NaN |
> | anGPT (no const. norm factor) |  2.667 (1.09%)   | 14.400 (2.93%) |
> | anGPT (no residual norm factor) |  2.719 (3.04%)   | 15.156 (8.33%) |
> | anGPT (no norm factor at all) |  2.718 (3.02%)   | 15.148 (8.28%) |
> | anGPT (no LERP) |  2.688 (1.89%)   | 14.700 (5.08%) |
>
> For reference, we include the results on a half/double token budget.
>
> |  | validation loss | PPL |
> | :---- |-----: |----: |
> | GPT+ (half token budget) |  2.787	(4.11%)   | 16.231 (11.62%) |
> | anGPT (half token budget) |  2.755 (4.42%)   | 15.718 (12.36%) |
> | GPT+ (double token budget) |  2.593 (-3.13%)   | 13.371 (-8.05%) |
> | anGPT (double token budget) |  2.565 (-2.78%)   | 13.000 (-7.07%) |
>
>
> ____
>
> **Table 2 Downstream Evaluation**
>
> We perform downstream evaluation on the 1B model size trained on token budgets from 10B to 42B.
>
> | PIQA Acc   |   10 B |   21 B |   42 B |
> |:-------|-----:|-----:|----:|
> | GPT+  |  0.669 |  0.674 |  0.700 |
> | anGPT    |  0.680 |  0.702 |  0.718 |
>
>
>
> | ARC‑Easy Acc   |   10 B |   21 B |   42 B |
> |:-------|-----:|-----:|----:|
> | GPT+   |  0.508 |  0.532 |  0.561 |
> | anGPT   |  0.539 |  0.561 |  0.596 |
>
>
> | HellaSwag Acc  |   10 B |   21 B |   42 B |
> |:-------|-----:|-----:|----:|
> | GPT+   |  0.338 |  0.363 |  0.400 |
> | anGPT    |  0.354 |  0.386 |  0.413 |
>
>
>
> | LAMBADA PPL   |   10 B |   21 B |   42 B |
> |:-------|-----:|-----:|----:|
> | GPT+        | 36.915 | 24.932 | 16.726 |
> | anGPT   | 28.780 | 17.577 | 14.679 |
>
>
>
> | WikiText word PPL   |   10 B |   21 B |   42 B |
> |:-------|-----:|-----:|----:|
> | GPT+    | 22.648 | 21.661 | 17.621 |
> | anGPT      | 20.518 | 17.734 | 16.175 |
>
>
> | WinoGrande Acc   |   10 B |   21 B |   42 B |
> |:-------|-----:|-----:|----:|
> | GPT+     |  0.522 |  0.536 |  0.554 |
> | anGPT    |  0.530 |  0.559 |  0.563 |
>
>
> | MMLU Acc   |   10 B |   21 B |   42 B |
> |:-------|-----:|-----:|----:|
> | GPT+                  |  0.232 |  0.239 |  0.243 |
> | anGPT                 |  0.232 |  0.241 |  0.256 |
>
>
>
> ____
>
> [1] Loshchilov, Ilya, et al. "ngpt: Normalized transformer with representation learning on the hypersphere." 13th International Conference on Learning Representations (ICLR 2025) .
>
> [2] Franke, Jörg, et al. "Improving deep learning optimization through constrained parameter regularization." Advances in Neural Information Processing Systems 37 (NeurIPS 2024): 8984-9025.

---

> > ### Comment · Reviewer_SY5g · 2025-08-07
> > **Response to Rebuttal**
> >
> > Thank you for the detailed response and the additional experiments. I appreciate the ablations on normalization sensitivity, the clarification of theoretical motivations, and especially the newly added downstream evaluations.
> >
> > The downstream results are encouraging and suggest that the proposed approach may transfer beyond pretraining loss. However, the evaluations are still limited in scope, focusing on a single model size and selected benchmarks. The theoretical assumptions behind approximate normalization also remain only partially validated, and analysis of potential failure modes is limited.
> >
> > Overall, I find the method promising and the response helpful, but I would like to maintain my score.

---

> ### Author Response · Authors · 2025-08-07
>
> Thank you for your detailed review, your evaluation of our rebuttal and your positive assessment. We have performed additional downstream evaluations across more benchmarks (OpenBookQA, CommonsenseQA, TruthfulQA, SciQ, GSM8K, TriviaQA) and model scales (0.5B), with anGPT outperforming GPT+ on most tasks, which we detail in our next comment. Our key contribution remains the comprehensive scaling analysis across multiple model sizes and compute budgets—to our knowledge, no prior work on transformer architecture modifications has systematically derived empirical scaling laws demonstrating consistent benefits across scales. The derived scaling laws show anGPT maintains nearly identical scaling exponents to GPT+ while consistently achieving lower loss, providing strong evidence for robust architectural improvements.
>
> ____
>
> ### **Additional Downstream Evaluation on 1B**
>
> We perform additional downstream evaluation on the 1B model size trained on token budgets from 10B to 42B.
>
>
> | OpenBookQA Acc   |   10 B |   21 B |   42 B |
> |:----------------------|-------:|-------:|-------:|
> | GPT+                  |  0.198 |  0.210 |  0.234 |
> | anGPT                 |  0.202 |  0.202 |  0.234 |
>
>
> | CommonsenseQA Acc   |   10 B |   21 B |   42 B |
> |:----------------------|-------:|-------:|-------:|
> | GPT+                  |  0.194 |  0.194 |  0.201 |
> | anGPT                 |  0.199 |  0.200 |  0.179 |
>
>
> | TruthfulQA MC1 Acc   |   10 B |   21 B |   42 B |
> |:----------------------|-------:|-------:|-------:|
> | GPT+                  |  0.229 |  0.240 |  0.218 |
> | anGPT                 |  0.240 |  0.235 |  0.220 |
>
>
> | TruthfulQA MC2 Acc  |   10 B |   21 B |   42 B |
> |:----------------------|-------:|-------:|-------:|
> | GPT+                  |  0.393 |  0.391 |  0.376 |
> | anGPT                 |  0.414 |  0.388 |  0.377 |
>
>
> | SciQ Acc  |   10 B |   21 B |   42 B |
> |:----------------------|-------:|-------:|-------:|
> | GPT+                  |  0.769 |  0.823 |  0.855 |
> | anGPT                 |  0.797 |  0.850 |  0.863 |
>
>
> | GSM8K CoT Flexible EM   |   10 B |   21 B |   42 B |
> |:----------------------|-------:|-------:|-------:|
> | GPT+                  |  0.000 |  0.000 |  0.000 |
> | anGPT                 |  0.000 |  0.000 |  0.000 |
>
>
> | TriviaQA EM   |   10 B |   21 B |   42 B |
> |:----------------------|-------:|-------:|-------:|
> | GPT+                  |  0.000 |  0.000 |  0.000 |
> | anGPT                 |  0.000 |  0.000 |  0.000 |

---

> > ### Author Response · Authors · 2025-08-07
> >
> > ### **Additional Downstream Evaluation on 0.5B**
> >
> > We perform additional downstream evaluation on the 0.5B model size trained on token budgets from 10B to 73B.
> >
> > | PIQA Acc              |   10 B |   21 B |   73 B |
> > |:----------------------|-------:|-------:|-------:|
> > | GPT+                  |  0.641 |  0.668 |  0.638 |
> > | anGPT                 |  0.532 |  0.686 |  0.700 |
> >
> > | ARC-Easy Acc          |   10 B |   21 B |   73 B |
> > |:----------------------|-------:|-------:|-------:|
> > | GPT+                  |  0.468 |  0.520 |  0.498 |
> > | anGPT                 |  0.268 |  0.525 |  0.549 |
> >
> > | HellaSwag Acc         |   10 B |   21 B |   73 B |
> > |:----------------------|-------:|-------:|-------:|
> > | GPT+                  |  0.317 |  0.346 |  0.347 |
> > | anGPT                 |  0.253 |  0.355 |  0.380 |
> >
> > | LAMBADA PPL           |   10 B |   21 B |   73 B |
> > |:----------------------|-------:|-------:|-------:|
> > | GPT+                  | 64.148 | 40.031 |124.307 |
> > | anGPT                 |    - | 25.070 | 16.710 |
> >
> > | WikiText word PPL     |   10 B |   21 B |   73 B |
> > |:----------------------|-------:|-------:|-------:|
> > | GPT+                  | 30.380 | 22.074 | 27.810 |
> > | anGPT                 |    - | 20.196 | 18.024 |
> >
> > | WinoGrande Acc        |   10 B |   21 B |   73 B |
> > |:----------------------|-------:|-------:|-------:|
> > | GPT+                  |  0.518 |  0.542 |  0.535 |
> > | anGPT                 |  0.502 |  0.548 |  0.569 |
> >
> > | MMLU Acc              |   10 B |   21 B |   73 B |
> > |:----------------------|-------:|-------:|-------:|
> > | GPT+                  |  0.229 |  0.233 |  0.249 |
> > | anGPT                 |  0.234 |  0.237 |  0.241 |
> >
> > | OpenBookQA Acc        |   10 B |   21 B |   73 B |
> > |:----------------------|-------:|-------:|-------:|
> > | GPT+                  |  0.174 |  0.198 |  0.158 |
> > | anGPT                 |  0.130 |  0.202 |  0.220 |
> >
> > | CommonsenseQA Acc     |   10 B |   21 B |   73 B |
> > |:----------------------|-------:|-------:|-------:|
> > | GPT+                  |  0.195 |  0.201 |  0.195 |
> > | anGPT                 |  0.194 |  0.190 |  0.190 |
> >
> > | TruthfulQA MC1 Acc    |   10 B |   21 B |   73 B |
> > |:----------------------|-------:|-------:|-------:|
> > | GPT+                  |  0.253 |  0.233 |  0.230 |
> > | anGPT                 |  0.621 |  0.250 |  0.240 |
> >
> > | TruthfulQA MC2 Acc    |   10 B |   21 B |   73 B |
> > |:----------------------|-------:|-------:|-------:|
> > | GPT+                  |  0.437 |  0.407 |  0.406 |
> > | anGPT                 |    - |  0.401 |  0.401 |
> >
> > | SciQ Acc              |   10 B |   21 B |   73 B |
> > |:----------------------|-------:|-------:|-------:|
> > | GPT+                  |  0.786 |  0.815 |  0.793 |
> > | anGPT                 |  0.148 |  0.831 |  0.855 |
> >
> > * The final checkpoint for the 10B anGPT run is corrupt  and we used a late training checkpoint.

---

### Decision · Program_Chairs · 2025-09-17

**Decision:**

Accept (poster)

**Comment:**

This paper presents an approximately normalized Transformer (anGPT) that uses fixed scalar multipliers, motivated by the concentration of measure phenomenon, to accelerate convergence by up to 40% and eliminate the need for weight decay and learning rate warm-up. The primary strengths identified by reviewers are the significant, empirically validated training speedup at small scales and the simplification of the training pipeline. Initial weaknesses centered on the limited scope of evaluation beyond pretraining loss and the partial validation of the underlying theoretical assumptions during training. During rebuttal, the authors provided extensive downstream evaluations and sensitivity analyses that demonstrated the method's practical benefits, addressing the most critical concerns. Although questions about the theoretical grounding remain, the strong empirical results and the prospect of practical utility support accepting the paper.